# Learning Color Equivariant Representations

**Yulong Yang**[1,*,†]    **Felix O'Mahony**[2,*]    **Christine Allen-Blanchette**[1,†]
[1]Princeton University, Princeton, USA    [2]University of Oxford, Oxford, UK
`yulong.yang@princeton.edu, felixomahony@gmail.com,`
`ca15@princeton.edu`

## Abstract

In this paper, we introduce group convolutional neural networks (GCNNs) equivariant to color variation. GCNNs have been designed for a variety of geometric transformations from 2D and 3D rotation groups, to semi-groups such as scale. Despite the improved interpretability, accuracy and generalizability of these architectures, GCNNs have seen limited application in the context of perceptual quantities. Notably, the recent CEConv network uses a GCNN to achieve equivariance to hue transformations by convolving input images with a hue rotated RGB filter. However, this approach leads to invalid RGB values which break equivariance and degrade performance. We resolve these issues with a lifting layer that transforms the input image directly, thereby circumventing the issue of invalid RGB values and improving equivariance error by over three orders of magnitude. Moreover, we extend the notion of color equivariance to include equivariance to saturation and luminance shift. Our hue-, saturation-, luminance- and color-equivariant networks achieve strong generalization to out-of-distribution perceptual variations and improved sample efficiency over conventional architectures. We demonstrate the utility of our approach on synthetic and real world datasets where we consistently outperform competitive baselines.

## 1 Introduction

The tremendous progress of image classification in the last decade can be readily attributed to the development of deep convolutional neural networks (Krizhevsky et al., 2012; Simonyan & Zisserman, 2014; He et al., 2016). The highly nonlinear mapping and large parameter space does not lend itself to interpretation easily, but even in early networks, representations of geometry and color were observed and recognized for their importance (Krizhevsky et al., 2012; Lenc & Vedaldi, 2015). While a large body of literature has worked to improve the robustness of neural networks to geometric transformations (Bruna & Mallat, 2013; Cohen & Welling, 2016; Gilmer et al., 2017; Qi et al., 2017; Hinton et al., 2018; Esteves et al., 2018a; Greydanus et al., 2019; Zhong & Allen-Blanchette, 2025), improving their robustness to perceptual variation has garnered considerably less attention.

A commonly used heuristic for improving network robustness to color variation is to perform mean subtraction and normalization on training set examples. This approach can work well when the training and testing datasets are drawn from the same distribution; however, for data that are collected at different points in time, or with different sensors, this is not likely to be the case. Consider, for example, the case of medical imaging where images of tissue samples collected from different labs (or from the same lab at different points in time) may have different characteristics due to variability in data collection protocols or imaging processes (Veta et al., 2016). This variability presents a significant challenge for convolutional neural networks which have been found to be sensitive to color variations even on the level of individual blocks (Engilberge et al., 2017). Moreover, when presented with color perturbations, conventional networks exhibit a significant drop in classification performance (De & Pedersen, 2021). One approach to mitigate the effect of color variation is to ignore color entirely by enforcing color invariance. This can be done by converting input images to grayscale, or enforcing representation similarity across color as is done in Pakzad et al. (2022). This approach has its own challenges, however, since in many domains, color is an important cue

---

* Equal contribution. † Corresponding author.

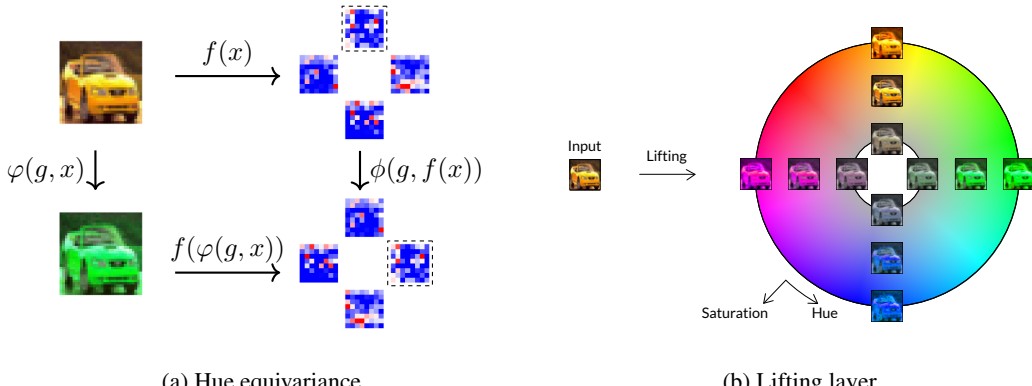

(a) Hue equivariance            (b) Lifting layer

Figure 1: **Color-equivariant network.** **(a)** The equivariance of our hue-equivariant model is illustrated by the commutativity of the (hue) rotation and neural network mapping. A hue rotation of 90° in the input image space (top-left to bottom-left), results in a feature map rotation at each layer of the network (top-right to bottom-right). Corresponding feature maps are highlighted with a blue border. **(b)** An input image (left) is lifted to the hue-saturation group (right) by shifting its hue and saturation values. For comparison, we illustrate the CEConv lifting layer in Appendix C.1.

for classification. Another approach is to use dataset augmentation, a technique that can improve robustness in the presence of known transformations (Benton et al., 2020). This approach, however, requires extended training times and does not provide improved interpretability.

In this work, we address the challenge of neural network sensitivity to color variation by leveraging the geometric structure of color in a group convolutional neural network. Geometric deep learning has gained strong interest in recent years due to its ability to capture information in interpretable and generalizable representations; by using a group convolutional neural network, we inherit these characteristics for the context of color variation. We represent color in the hue saturation luminance (HSL) color space, and leverage the insight that the hue, saturation, and luminance can be modeled with geometric group structure. Using this approach, our network retains color information in a structured representation throughout the processing pipeline, thereby allowing color information to be used and/or discarded intentionally.

Recently, *Color Equivariant Convolutional Networks* (Lengyel et al., 2024), proposed a group convolutional neural network architecture for hue-equivariant representation learning. Most notably, CEConv identifies hue transformations with the 2D rotation group. However, the proposed lifting layer introduces invalid RGB values which break equivariance, and degrade performance. To resolve this issue, our lifting layer operates on the input image instead of the network filters, which produces an equivariant descriptor without projection induced artifacts. This seemingly minor change improves equivariance error by more than three orders of magnitude, and stabilizes network performance across discretizations. We also expand the notion of color equivariance to capture variation in saturation and luminance. Specifically, we identify variations in saturation and luminance with the 1D translation group. With this identification, we propose a group-equivariant network where variations in hue, saturation and luminance are represented as a geometric transformations.

## 2 RELATED WORK

**Group Convolutional Neural Networks.** The generalization performance of convolutional neural networks on image processing tasks is attributed, in part, to the equivariance of planar convolution to 2D translations. This insight has garnered considerable attention and led to a strong interest in the development of convolutional neural networks equivariant to other symmetry groups (Kondor & Trivedi, 2018; Cohen et al., 2019b). Previous works introduce group convolutional networks for finite and continuous groups. A framework for finite group convolution was developed in Cohen & Welling (2016), and demonstrated for the 2D rotation and reflection groups. The authors achieve equivariance to specific symmetry groups by convolving input images and feature maps with the group orbit of a learned filter bank. In Worrall et al. (2017), the authors design for equivariance to the continuous 2D rotation group by constraining filter representations with circular harmonic structure.

Group convolutional networks have also been designed for use in the context of scale symmetry. In Esteves et al. (2018b), the authors represent input images in log-polar coordinates where 2D rotation and scale transformations present as 2D translations. To navigate the semi-group structure of scale transformations, the authors in Worrall & Welling (2019) approximate the scale space as finite and use dilated convolutions in a group equivariant architecture. The SREN network proposed in Sun & Blu (2022) designs for equivariance to the continuous 2D rotation and isotropic scaling group by constraining filter representations to be the linear combination of a windowed Fourier basis.

Group convolutional networks have also been designed for groups acting in higher dimensions. For 3D rotational symmetry, Thomas et al. (2018) and Esteves et al. (2019) use a spherical harmonics based representation with 3D point clouds and spherical image inputs respectively; and Batzner et al. (2022) introduce an equivariant network for transformations of the Euclidean group in 3D. In Finzi et al. (2020), the authors introduce a framework for general Lie groups whose exponential map is surjective, and Cohen et al. (2019a) introduces a framework for more general manifolds. By designing for known symmetries in the task, these works provide improved interpretability, training efficiency and generalizability over conventional convolutional networks.

Other group equivariance methods encourage transformation equivariance using a soft penalty term on representation dissimilarity (Lenc & Vedaldi, 2015; Gupta et al., 2023). These methods are separate from the group convolutional neural network literature, and different from what we propose. Our model leverages the geometric structure of hue, saturation and luminance to design color-equivariant networks, bringing the benefits of group-equivariant networks to perceptual transformations.

**Color Invariance.** Several prior works attempt to mitigate the effect of color variation in image processing tasks, for example, in Chong et al. (2008), the authors represent images in a color space where pixel values are invariant to changes in luminance for luminance invariant image segmentation, and in Pakzad et al. (2022), the authors penalize changes in their latent representations due to color variation for color invariant skin lesion identification. While color invariance resolves the effect of color variation, color has been identified as an important cue in representation learning (Engilberge et al., 2017; De & Pedersen, 2021) which is discordant with the goal of color invariance. In contrast to these approaches, our work leverages the geometric structure of hue, saturation and luminance to construct a convolutional neural network equivariant to variations in these quantities by design.

Most similar to ours is the model proposed in Lengyel et al. (2024), a hue-equivariant network for improved performance in the presence of color variation. Our color-equivariant networks differ from the work proposed in Lengyel et al. (2024) in two important ways. First, we lift the input image instead of the filters of the first layer which circumvents the issue of invalid hue rotations suffered by the network in Lengyel et al. (2024); and second, we expand the notion of color equivariance by introducing networks that are equivariant to hue, saturation, and luminance.

## 3 PRELIMINARIES

In this section we describe our notation, and review definitions of the group action, equivariance, and group convolution.

**Notation.** We use $f^l$ to denote the $l$-th feature map ($f^0$ to denote an input image), and $f_j^l$ to denote the $j$-th channel of that feature map. We use $\psi_i^l$ to denote the $i$-th filter of the $l$-th layer, and $\psi_{i,j}^l$ to denote the $j$-th channel of that filter.

**Group action.** Adapted from Gallier & Quaintance (2020). Given a set $X$ and a group $G$, the action of the group $G$ on $X$ is a function $\varphi : G \times X \to X$ satisfying the following:

1. For all $g, h \in G$ and all $x \in X$, $\varphi(g, \varphi(h, x)) = \varphi(gh, x)$

2. For all $x \in X$, $\varphi(1, x) = x$ where $1 \in G$ is the identity element of $G$.

The set $X$ is called a (left) $G$-set.

**Equivariance.** Adapted from Gallier & Quaintance (2020). Given two $G$-sets $X$ and $Y$, and group actions $\varphi : G \times X \to X$ and $\phi : G \times Y \to Y$, a function $f : X \to Y$ is said to be equivariant, if and only if for all $x \in X$, and $g \in G$, $f(\varphi(g, x)) = \phi(g, f(x))$. When $f$ is invariant to the action of

$G$, the group action $\phi$ is the identity map and we write $f(\varphi(g, x)) = f(x)$. The equivariance of our network to hue shifts is illustrated in Figure 1a.

**Group convolution.** In a conventional CNN, the input to convolution layer $l$, denoted $f^l : \mathbb{Z}^2 \to \mathbb{R}^{K^l}$, is convolved with a set of $K^{l+1}$ filters, denoted $\psi_i^l : \mathbb{Z}^2 \to \mathbb{R}^{K^l}$, where $i$ ranges from 1 to $K^{l+1}$. The result of the convolution can be written:

$$f_i^{l+1} = [f^l * \psi_i^l](x) = \sum_{y \in \mathbb{Z}^2} \sum_{k=1}^{K^l} f_k^l(y) \psi_{i,k}^l(x - y), \ x \in \mathbb{Z}^2. \tag{1}$$

The convolution of conventional CNNs is equivariant to the action of the group $(\mathbb{Z}^2, +)$, that is, the group formed by summing over the integers. The more general group convolution can be written:

$$[f^l * \psi_i^l](g) = \sum_{h \in G} \sum_{k=1}^{K^l} f_k^l(h) \psi_{i,k}^l(h^{-1} g), \tag{2}$$

and is equivariant to the action of the group $G$.

## 4 METHOD

In this section we present our color-equivariant network. We begin by presenting definitions for the hue and saturation groups and their group actions, then define the lifting layers and group convolution layers for our color-equivariant networks.

**Hue group and group action.** In the HSL color space, hue is represented by angular position, and can therefore be identified with the 2D rotation group. As in group convolutional network (Cohen & Welling, 2016), we consider a finite group representation. Specifically, we identify elements of the discretized hue group, $H_N$, where the subscript $N$ indicates the order of (i.e. the number of elements in) the group, with those of the cyclic group $C_N$. We prove $H_N$ is a group in Appendix A.1.

We define the action of hue group on HSL images, $x \in X$ where $x : \mathbb{Z}^2 \to \mathbb{R}^3$, and functions on the discrete hue group, $y \in Y$, where $y : \mathbb{Z}^2 \times H_N \to \mathbb{R}^K$. An element of the hue group acts on an HSL image by the group action $\varphi_h : H_N \times X \to X$, which shifts the hue channel of the image. Concretely, for an HSL image $x \in X$ defined as the concatenation of hue, saturation and luminance channels, i.e., $x = (x_h, x_s, x_l)$, the action of an element $h_i$ of the hue group $H_N$ is given by

$$\varphi_h(h_i, x) = ((x_h + h_i)(\bmod c), x_s, x_l), \tag{3}$$

where $(\cdot) \bmod (\cdot)$ denotes the modulus operation and is applied to the pixel value which ranges from 0 to $c$.

An element of the hue group acts on a function on the discrete hue group by the group action $\phi_h : H_N \times Y \to Y$, which "rotates" the function on the group. Concretely, for a function $f$ on the discrete hue group $H_N$ defined as the concatenation of functions $f = (f_1, \ldots, f_N)$, the action of an element $h_i$ in the hue group $H_N$ is given by

$$\phi_h(h_i, f) = (f_{(1+i)(\bmod N)}, \ldots, f_{(N+i)(\bmod N)}). \tag{4}$$

The action of $\varphi_h$ on an input image, and $\phi_h$ on a feature map are shown in Figure 6a. We prove $\varphi_h$ and $\phi_h$ are group actions in Appendix B.1.

**Saturation group and group action.** We expand the notion of color equivariance proposed in Lengyel et al. (2024) to include equivariance to saturation shifts. In the HSL color space, saturation can be represented by a real number in the interval $[0, 1]$; we introduce two approximations to give the saturation space group structure. First, we observe that there is a bijection between the real numbers and the open interval $(0, 1)$, so we can use the structure of the group $(\mathbb{R}, +)$. Second, we consider a finite subset of the group, a necessary and commonly used practice in both conventional CNNs (recall that the translation group is infinite) and group-equivariant CNNs such as Worrall & Welling (2019). With these approximations, the discretized saturation group $S_N$ is isomorphic to the integers with addition, $(\mathbb{Z}, +)$. We prove $S_N$ is a group in Appendix A.2.

We define the action of the saturation group on HSL images, $x \in X$ where $x : \mathbb{Z}^2 \to \mathbb{R}^3$, and functions on the discrete saturation group, $y \in Y$, where $y : \mathbb{Z}^2 \times S_N \to \mathbb{R}^K$. An element of the

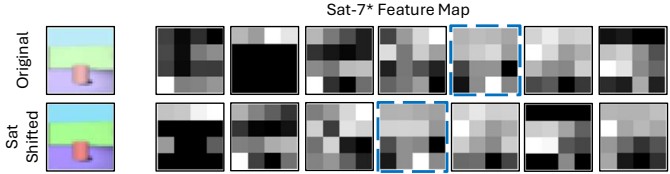

Figure 2: **Saturation-equivariant feature maps.** We illustrate the equivariance of our saturation-equivariant model. A saturation shift in the input image space (top-left to bottom-left), results in a feature map translation at each layer of the network (top-right to bottom-right). Corresponding feature maps are highlighted with a blue border.

saturation group acts on an HSL image by the group action $\varphi_s : S_N \times X \to X$, which shifts the saturation channel of the image. Concretely, for an HSL image $x \in X$ defined as the concatenation of hue, saturation and luminance channels, i.e., $x = (x_h, x_s, x_l)$, the action of an element $s_i$ of the saturation group $S_N$ is given by

$$\varphi_s(s_i, x) = (x_h, \min(x_s + s_i, c), x_l). \tag{5}$$

An element of the saturation group acts on a function on the discrete saturation group by the group action $\phi_s : S_N \times Y \to Y$, which "translates" the function on the group. Concretely, for a function $f$ on the discrete saturation group $S_N$ defined as the concatenation of functions $f = (f_1, \ldots, f_N)$, the action of an element $s_i$ in the saturation group $S_N$ is given by

$$\phi_s(s_i, f) = (f_{1+i}, \ldots, f_N, \underbrace{\mathbf{0}, \ldots, \mathbf{0}}_{i}). \tag{6}$$

The action of $\varphi_s$ on an input image, and $\phi_s$ on a feature map are shown in Figure 2. We prove $\varphi_s$ and $\phi_s$ are group actions in Appendix B.2. We define the luminance group and group action similarly (see Appendix F.1).

**Hue-Saturation group action.** We define the action of the hue-saturation group as a composition of the hue and saturation group actions. An element of the hue-saturation group acts on an HSL image by the group action $\varphi_{hs} : H_N \times S_M \times X \to X$, which shifts both the hue and saturation channels of the image. For an HSL image $x \in X$ defined as the concatenation of hue, saturation and luminance channels, i.e., $x = (x_h, x_s, x_l)$, the action of an element $(h_i, s_j)$ of the hue-saturation group $H_N \times S_M$ is given by

$$\varphi_{hs}((h_i, s_j), x) = \varphi_h(h_i, \varphi_s(s_j, x)). \tag{7}$$

An element of the hue-saturation group acts on a function on the discrete saturation group by the group action $\phi_{hs} : H_N \times S_M \times Y \to Y$, which "rotates" and "translates" the function on the group. Concretely, for a function $f$ on the discrete hue-saturation group $H_N \times S_M$ defined as the concatenation of functions $f = (f_{11}, \ldots, f_{1M}, f_{21}, \ldots, f_{NM})$, the action of an element $(h_i, s_j)$ in the hue-saturation group $H_N \times S_M$ is given by

$$\phi_{hs}((h_i, s_j), f) = \phi_h(h_i, \phi_s(s_j, f)). \tag{8}$$

We prove $\varphi_{hs}$ and $\phi_{hs}$ are group actions in Appendix B.

**Lifting layer.** The first layer of a group convolutional neural network "lifts" the input image to the group (see Figure 1b). We can lift an input image to the product space of the image grid $\mathbb{Z}^2$, and discretized hue shifts $H_N = \{h_0, h_1, \ldots, h_N\}$, by convolving with hue shifted filters,

$$[f^0 * \psi_i^0](g_{x,j}) = \sum_{y \in \mathbb{Z}^2} \sum_{k=1}^{K^0} f_k^0(y) h_j \psi_{i,k}^0(x - y). \tag{9}$$

Here we denote an element of the product space $g_{x,i}$, where the subscript $x$ references an element of $\mathbb{Z}^2$, and $j$ references the element $h_j$ in $H_N$. The input image $f^0$ and lifting filters $\psi_{m,n}$ are functions on $\mathbb{Z}^2$. This is the approach taken in CEonv (Lengyel et al., 2024). The authors shift the hue of a filter in the RGB space by rotating its values about an axis passing through the point $p = (1, 1, 1)$ as shown

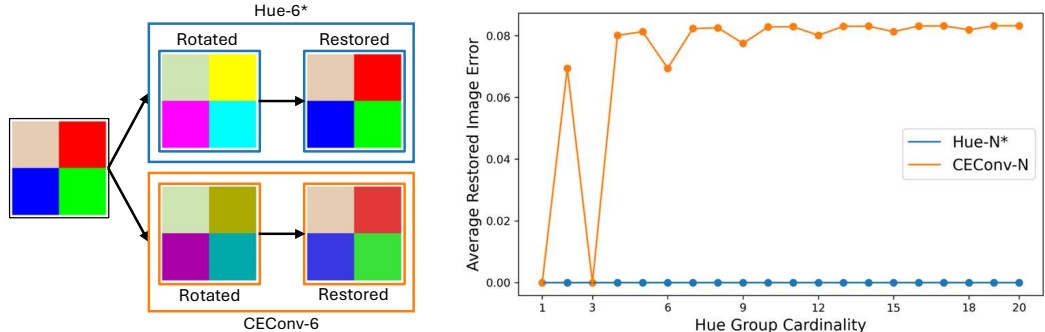

Figure 3: **Impact of order on hue rotation invertibility.** Our lifting layer (blue) operates on HSL input images where each hue rotation is invertible. The lifting layer proposed in CEConv (orange) operates on RGB filters and suffers from invalid hue rotations for all discretizations of the hue group except for $N = 1$ and $N = 3$ (i.e., symmetries of the axis-aligned RGB cube). **(Left)** We show the impact of invalid hue rotations on a four pixel image. We rotate the image $60°$ using our proposed lifting layer (top-left) and the CEConv lifting layer (bottom-left). Subsequently applying a $-60°$ rotation yields an image that is indistinguishable from the original using our approach (top-left), and one with visible artifacts using the CEConv approach (bottom-left). **(Right)** We show the average restored image error for both approaches. Our approach results in a consistently negligible restored image error, however, the CEConv approach results in a restored image error exceeding $7\%$ for all discretizations of the hue group except $N = 1$ and $N = 3$.

in Figure 8b in Appendix C.1. As described in Lengyel et al. (2024), this approach results in invalid hue rotations for all discretizations of the hue group that are not symmetries of the axis-aligned RGB cube (i.e., $N = 1$ and $N = 3$). To remedy this, the authors project invalid RGB values to the nearest point on the RGB cube. However, this projection breaks invertibility of the hue shift (see Figure 3), yielding a descriptor that is only approximately hue equivariant.

Lifting to the group can alternatively be achieved by transforming the input image rather than the filters,

$$f^1(g_{x,j}) = \varphi_h(h_j, f^0)(x).$$ (10)

We use this approach in our implementation as it circumvents the issue of invalid hue rotations, producing a reliably equivariant descriptor. The lifting layers for the saturation group, luminance group, hue-saturation group, and hue-luminance group are constructed analogously.

**Equivariance of the lifting layer.** We can show that our lifting layer is equivariant to hue shifts. We let $f^1(g_{x,j}) = \varphi_h(h_j, f^0)(x)$ as in Equation (10) and show that a hue shift of $f^0$ results in a hue shift of $f^1$. In the first step we use the fact that $\varphi_h$ is a group action (see Appendix B.1), and in the second step we use commutativity of the hue group:

$$\varphi_h(h_j, \varphi_h(h_m, f^0))(x) = \varphi_h(h_j h_m, f^0)(x)$$ (11)

$$= \varphi_h(h_m h_j, f^0)(x)$$ (12)

$$= \varphi_h(h_m, \varphi_h(h_j, f^0))(x)$$ (13)

$$= \varphi_h(h_m, f^1(g_{x,j})).$$ (14)

Equivariance of the saturation, luminance, hue-saturation, and hue-luminance lifting layers can be shown analogously.

**Group convolution layer.** For all layers after the first layer, the feature maps $f^l$ are on the product space $\mathbb{Z}^2 \times H_N$. Since hue shifts can only be performed on three dimensional inputs, we leverage the identification of hue shifts with the discrete 2D rotation group, i.e., $H_N \cong C_N$ and interpret the feature maps $f^l$ as functions on the product space $G := \mathbb{Z}^2 \times C_N$. With this interpretation we perform group convolution on $G$ as follows,

$$[f^l * \psi_i^l](g) = \sum_{h \in H} \sum_{k=1}^{K^l} f_k^l(h) \psi_{i,k}^l(h^{-1}g),$$ (15)

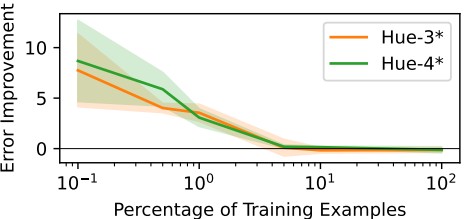

Figure 4: **Model sample efficiency.** We show the error improvement (higher is better) over the Z2CNN baseline as a function of the percentage of training examples used. The advantage of our Hue-$N$ models increases as the percentage of training examples used decreases.

Table 1: **Generalization to global hue-shift.** Classification error on the Hue-shift MNIST dataset is reported. Our hue-equivariant models (Hue-$N$) perform comparably to baselines on the in-distribution case ($A/A$), and outperform baselines on the out-of-distribution case ($A/B$).

| Network | $A/A$ | $A/B$ | Params |
|---|---|---|---|
| Z2CNN | **1.54 (0.10)** | 57.38 (22.06) | 22,130 |
| Hue-3* | 1.79 (0.25) | **1.81 (0.29)** | 22,658 |
| Hue-4* | 1.97 (0.25) | 2.08 (0.15) | 25,690 |
| CEConv-3 | 1.79 (0.13) | 1.83 (0.19) | 28,739 |
| CEConv-4 | 3.04 (0.27) | 3.09 (0.25) | 30,539 |

where $\psi^l$ is a function on $G$. Group convolution on the saturation, luminance, and hue-saturation-luminance groups are performed analogously.

## 5 EXPERIMENTS

In this section, we highlight the sample efficiency of our model in the context of hue generalization; the stability of our hue-equivariant representations compared to CEConv (Lengyel et al., 2024); the utility of a notion of color equivariance that includes saturation and luminance; and the utility of our representations for color-based sorting. Our models achieve strong performance on extensive experiments, against competitive baselines. Additional experiments are presented in Appendix F.

### 5.1 HUE-SHIFT MNIST CLASSIFICATION

We demonstrate improved generalization to global hue-shifts and higher sample efficiency compared to a conventional CNN model with a similar number of parameters on our Hue-shift MNIST dataset. Our dataset is a variation of MNIST (LeCun et al., 1998) with 60k training examples and 10k test examples classified into one of 10 categories. Additional details are provided in Appendix D.1.

**Generalization to global hue-shift.** The generalization performance of our hue-equivariant models, Hue-$N$ ($N$ indicates the order of the discrete hue group), and a conventional CNN model, Z2CNN (Cohen & Welling, 2016), are reported in Table 1. On the in-distribution test case ($A/A$), the performance of our model and the conventional CNN model are comparable. However, on the out-of-distribution test case ($A/B$), the performance of our model is preserved, while the performance of the conventional CNN model deteriorates significantly.

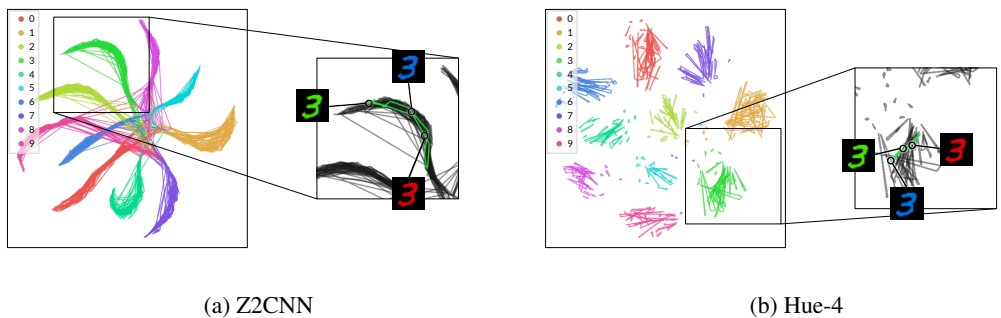

(a) Z2CNN

(b) Hue-4

Figure 5: **Hue-shift MNIST feature map visualization.** We compare the feature map trajectories of MNIST digits as their hue is varied from $1°$ to $360°$. The color of the trajectory corresponds to the class label. **(a)** tSNE projection of hue shifted feature map trajectories in the Z2CNN baseline model. As the hue of the input changes, the location of the digit in the feature space changes significantly. **(b)** tSNE projection of hue shifted feature map trajectories in our hue-equivariant CNN. In contrast to the Z2CNN baseline, the location of the digit in the feature space changes minimally.

Table 2: **Generalization to local hue-shift.** Classification error on the 3D Shapes dataset is reported. Our model (Hue-$N$) and CEConv-$N$ achieve improved generalization performance over the conventional CNN model (Z2CNN). Our approach is robust to the choice of hue group discretization, while the performance of CEConv deteriorates when $N$ is not a symmetry of the axis-aligned RGB cube.

| Network | $A/A$ | $A/B$ | $A/C$ | Params |
|---------|-------|-------|-------|--------|
| Z2CNN | 0.00 (0.00) | 51.25 (9.59) | 26.66 (19.60) | 20,192 |
| Hue-3* | 0.00 (0.00) | 0.03 (0.04) | 0.04 (0.03) | 19,478 |
| Hue-4* | **0.00 (0.00)** | **0.00 (0.00)** | **0.00 (0.00)** | 21,832 |
| CEConv-3 | 0.00 (0.00) | 0.02 (0.03) | 0.05 (0.04) | 26,441 |
| CEConv-4 | 0.00 (0.03) | 0.60 (0.25) | 0.53 (0.43) | 30,792 |

The performance gap on the out-of-distribution test case can be attributed to the difference in internal representation structure. To understand this better, we generate feature representation trajectories in the penultimate layer of our network and the baseline architecture by continuously varying the hue of an input image. We then visualize these trajectories using tSNE (van der Maaten & Hinton, 2008) projection (see Figure 5). For the baseline architecture, the trajectories of different digits overlap for some hues (Figure 5a), but for our model, the trajectories of different digits are confined to separate clusters (Figure 5b).

**Model sample efficiency.** The sample efficiency of our hue-equivariant models (Hue-$N$) and a conventional CNN model (Z2CNN) are reported in Figure 4. The advantage of our hue-equivariant models increases as the percentage of the training set used decreases, illustrating improved sample efficiency (Elesedy, 2022). The error improvement is defined as the difference between the performance of the proposed model and the performance of the baseline (Z2CNN).

## 5.2 HUE-SHIFT 3D SHAPES CLASSIFICATION

We demonstrate improved generalization to local hue-shifts and significantly reduced equivariance error compared to CEConv on the 3D Shapes classification dataset (Burgess & Kim, 2018). The 3D Shapes dataset consists of RGB images of 3D shapes, where the color of the shape, the floor, and the walls vary across examples, as does the scale and orientation of the shape. There are 48k examples in the training set, and 12k examples in each of the test sets. Additional details are provided in Appendix D.2.

**Generalization to local hue-shift.** The generalization performance of our hue-equivariant models, Hue-$N$ ($N$ indicates the cardinality of the discrete hue group), a conventional CNN model, Z2CNN (Cohen & Welling, 2016), and the CEConv-$N$ models are reported in Table 2. Our Hue-4 model outperforms all other models on the in-distribution ($A/A$), global hue-shift out-of-distribution ($A/B$) and local hue-shift out-of-distribution ($A/C$) test sets. The performance of CEConv-4 is significantly worse than all other hue-equivariant models. We attribute the relatively poor generalization performance of the CEConv-4 model to the feature map equivariance error.

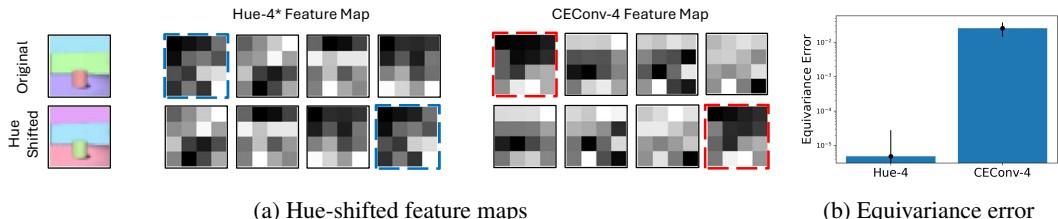

(a) Hue-shifted feature maps                    (b) Equivariance error

Figure 6: **Hue-equivariant feature maps.** (a) We qualitatively compare hue-shifted feature maps produced using our lifting layer and the CEConv lifting layer. Highlighted feature maps obtained using our lifting layer (Hue-4) are qualitatively indistinguishable, while there are visible discrepancies in the feature maps of CEConv-4. (b) We quantitatively compare the equivariance-error of the hue-shifted feature maps for each approach. The equivariance error resulting from our lifting layer (Hue-4) is more than three orders of magnitude lower than the error resulting from CEConv-4.

Table 3: **Generalization to saturation-shift.** Classification error on the Camelyon17 dataset is reported. Our model (Sat-$N$ and Hue-$M$-Sat-$N$) achieves improved generalization performance over the conventional CNN model (ResNet50).

| Networks | Error | Param |
|---|---|---|
| ResNet50 | 28.91 (7.58) | 23.5M |
| Sat-3* | **16.08 (2.68)** | 23.3M |
| Hue-4-Sat-3* | 19.06 (4.92) | 23.0M |
| CEConv-3 | 28.76 (9.93) | 23.1M |

**Lifting layer equivariance error.** The lifting layer equivariance error of our Hue-4 model and the CEConv-4 model are reported in Figure 6b. As discussed in Section 4, our lifting layer operates on HSL images and does not induce the invalid hue rotations observed in CEConv. We quantitatively assess how lifting layer invertibility impacts descriptor equivariance following Zhdanov et al. (2024). We compute the equivariance error as the relative equivariance,

$$\text{Equivariance Error} = \frac{|f(\varphi_h(h_i, x)) - \phi_h(h_i, f(x))|}{|f(\varphi_h(h_i, x)) + \phi_h(h_i, f(x))|} \tag{16}$$

and show that the equivariance error of CEConv-4 is more than three orders of magnitude higher than our Hue-4 model (see Figure 6b).

### 5.3 Camelyon17 Classification: Saturation shift in the wild

We demonstrate improved generalization to saturation-shifts compared to ResNet50 (He et al., 2016) and CEConv-3 on the Camelyon17 classification dataset (Bandi et al., 2018). The Camelyon17 dataset consists of images of human tissue collected from five different hospitals. Variation in tissue images results from variation in the data collection and processing procedures. The dataset consists of 387,490 examples, 302,436 of which are used for training and 85,054 of which are used for testing. Hospitals 1, 2, and 3 are represented in the training set, and hospital 5 is represented in the testing set. Additional details are provided in Appendix E.3.

**Generalization to saturation-shift.** We expand the notion of color equivariance proposed in Lengyel et al. (2024) to capture variation in saturation. The generalization performance of our saturation-equivariant models, Resnet50 (He et al., 2016), and CEConv-3 are reported in Table 3. Our saturation- and color-equivariant models outperform all other models, while CEConv performs comparably to the conventional CNN model.

### 5.4 Color shift in the wild

We demonstrate improved generalization to global color-shifts compared to ResNet (He et al., 2016) and CEConv on the Caltech-101 (Li et al., 2022), Oxford-IIT Pets (Parkhi et al.), Stanford

Table 4: **Generalization to color-shift.** Classification error on the Caltech-101, Oxford-IIT Pets, Stanford Cars, CIFAR-10, CIFAR-100, STL-10 datasets are reported. Our models (Hue-$N$, Sat-$N$ and Hue-$M$-Sat-$N$) outperform CEConv-$N$ models on all datasets, and are competitive with ResNet. We use ResNet-aug and Resnet-gray to denote a ResNet architecture trained on a color-jitter augmented dataset and grayscale image dataset respectively.

| Network | Caltech 101 | CIFAR-10 | CIFAR-100 | Stanford Cars | Oxford Pets | STL-10 |
|---|---|---|---|---|---|---|
| ResNet | 32.68 (1.55) | **7.86 (1.14)** | **32.00 (0.63)** | 25.41 (0.96) | 31.52 (2.05) | 18.59 (1.65) |
| ResNet-gray | 33.79 (3.09) | 8.45 (0.68) | 32.04 (0.66) | 24.71 (0.93) | 30.38 (0.35) | 18.71 (1.47) |
| ResNet-aug | 32.90 (0.82) | 8.33 (0.44) | 32.27 (0.18) | 22.38 (1.65) | 30.06 (0.52) | **17.89 (1.48)** |
| Hue-3* | 34.32 (0.34) | 8.41 (0.39) | 33.28 (0.63) | 22.19 (2.20) | 31.37 (3.50) | 19.82 (0.98) |
| Hue-4* | **32.23 (1.07)** | 8.83 (0.64) | 34.70 (0.89) | **20.38 (1.06)** | **27.39 (0.68)** | 20.73 (1.11) |
| Hue-4-Sat-3* | 38.14 (1.07) | 10.68 (0.78) | 33.27 (0.31) | 24.79 (3.87) | 29.84 (1.34) | 20.53 (0.73) |
| Sat-3* | 41.64 (1.40) | 9.24 (0.27) | 39.33 (0.45) | 31.90 (10.03) | 36.87 (5.57) | 20.71 (1.10) |
| CEConv-3 | 34.74 (0.83) | 8.86 (0.33) | 34.95 (0.44) | 23.97 (1.56) | 31.08 (2.54) | 24.29 (1.31) |
| CEConv-4 | 33.52 (0.48) | 9.28 (0.24) | 35.46 (0.35) | 24.08 (0.66) | 33.70 (1.50) | 21.90 (1.64) |

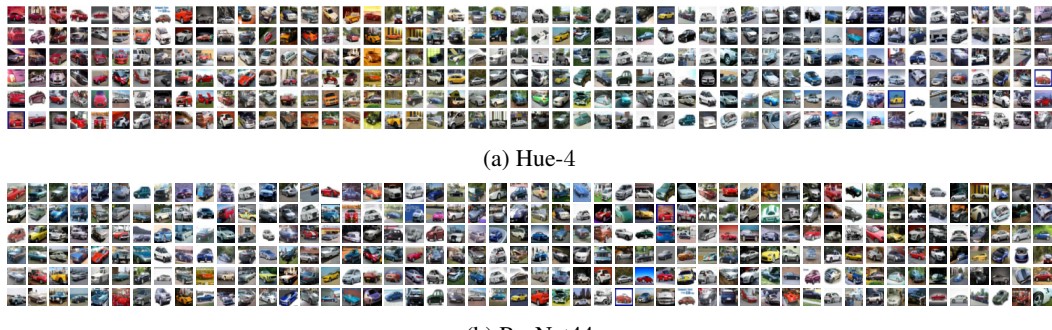

(a) Hue-4

(b) ResNet44

Figure 7: **Color sorting on CIFAR-10.** We show images from CIFAR-10 automobile class ordered by pairwise distance using Hue-4 and ResNet44 feature maps. The structure of the Hue-4 feature maps naturally allow for color based sorting, whereas the ResNet44 feature maps do not. We include visualizations of the entire automobile class sorted using Hue-$N$ and CEConv-$N$ in Appendix C.3.

Cars (Krause et al., 2013), STL-10 (Coates et al., 2011), CIFAR-10 and CIFAR-100 (Krizhevsky et al., 2009) datasets. We report results on the ImageNet dataset in Appendix F.2 and provide additional details for all experiments in Appendix E.

**Generalization to color-shift.** The generalization performance of our hue-equivariant models, Hue-$N$, saturation-equivariant model, Sat-$N$, color-equivariant models Hue-$M$-Sat-$N$, ResNet44 model, and the CEConv-$N$ models are reported in Table 4. Notably, our color-equivariant network outperforms baseline models on all fine-grained classification tasks (i.e., Stanford Cars and Oxford-IIT Pets datasets). We attribute the performance gap between our Hue-4 model and the CEConv-4 model to the significant difference in their equivariance error.

**Color sorting on CIFAR-10.** While our hue-equivariant model performs on-par with the conventional CNN model on this dataset, the structure of our representation can be leveraged for tasks that are not possible with conventional architectures alone. To illustrate this, we use our model to sort images in the automobile class by hue (see Figure 7a). To produce an hue ordering, we compute the hue-shifted pairwise Euclidean distance between hue group feature maps of instances in the automobile class. Images $x_1$ and $x_2$ are close in the hue space if

$$\arg\min_i \|\phi_h(h_i, \Psi(x_1)), \Psi(x_2)\| = 0, \tag{17}$$

where $\Psi(x_j)$ denotes the penultimate feature map for image $j$ and $h_i$ is the $i$-th element of the hue group. Sorting using the penultimate feature map of the ResNet44 model is shown in Figure 7b.

## 6 CONCLUSION

In this paper we address the challenge of learning hue-, saturation- and luminance-equivariant representations. Leveraging the observation that these perceptual transformations have geometric structure, we propose a group structure for each, and a group convolutional neural network that is equivariant to these transformations by design. By encoding perceptual variation in a finite group structure, we achieve strong generalization performance and sample efficiency and can use learned representations for tasks such as color-based sorting.

**Limitations and future work.** GCNNs are more computationally expensive than their conventional counterparts since they require computation of the filter orbit at each layer. Concretely, the computational cost of GCNNs is approximately equal to that of conventional CNNs with a filter bank size equal to that of the augmented filter bank used in GCNNs (Cohen & Welling, 2016). GCNNs also require a finite filter orbit which can at best approximate a continuous group. Future work will investigate a continuous representation of the group to address both of these limitations.

### ACKNOWLEDGMENTS

We sincerely thank the ICLR 2025 reviewers and area chair for their thoughtful feedback; the discussions and additional experiments have meaningfully improved our paper. We thank Minseok Kim for their insights and discussion.

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

## APPENDIX

### A    HUE AND SATURATION GROUP

In this section we prove the proposed discretized hue group and saturation group satisfy the axioms for a group. By definition, a group is a non-empty set $G$ along with a binary operation on $G$ that satisfies:

1. Closure. For all $a, b \in G$,

$$a \cdot b \in G. \tag{18}$$

2. Associativity. For all $a, b, c \in G$,

$$(a \cdot b) \cdot c = a \cdot (b \cdot c). \tag{19}$$

3. Identity element. There exists an element $1 \in G$ so that for all $a \in G$,

$$a \cdot 1 = 1 \cdot a = a. \tag{20}$$

4. Inverse element. For each $a \in G$, there exists $a^{-1} \in G$ so that,

$$a \cdot a^{-1} = a^{-1} \cdot a = 1. \tag{21}$$

## A.1 HUE GROUP

We show the set $H_N$,

$$H_N = \left\{ 0, \ldots, {}^{2\pi k}/_N, \ldots, {}^{2\pi(N-1)}/_N \right\}, \tag{22}$$

where $0 \le k < N$, together with the binary operation, $\cdot : H_N \times H_N \to H_N$, where

$$a \cdot b \mapsto (a + b) \,(\mathrm{mod}\, 2\pi) \tag{23}$$

for any $a, b \in H_N$, is a group. To do this, we prove the group axioms (18)-(21).

**Closure** is satisfied if for any $a, b \in H_N$, we have $a \cdot b \in H_N$. Any $a, b \in H_N$ can be expressed $a = {}^{2\pi k_a}/_N$ and $b = {}^{2\pi k_b}/_N$ for some integers $k_a$ and $k_b$ in the range $0 \le k_a, k_b < N$. The product of $a$ and $b$ can then be written

$$a \cdot b = \left( \frac{2\pi (k_a + k_b)}{N} \right) (\mathrm{mod}\, 2\pi). \tag{24}$$

To prove closure, we consider two cases: $k_a + k_b < N$, and $k_a + k_b \ge N$. In the first case, that is, $k_a + k_b < N$, we immediately have

$$a \cdot b = \frac{2\pi (k_a + k_b)}{N} \in H_N, \tag{25}$$

since $k_a + k_b$ is in the range $0 \le k_a + k_b < N$. In the second case, that is, $k_a + k_b \ge N$, we can write $k_a + k_b$ in quotient remainder form,

$$k_m * N + k_r = k_a + k_b, \tag{26}$$

where $k_m$ is the divisor, and $k_r$ is the remainder. Using this form we can write the product of elements $a$ and $b$

$$a \cdot b = \left( \frac{2\pi (k_m * N + k_r)}{N} \right) (\mathrm{mod}\, 2\pi) \tag{27}$$

$$= \left( 2\pi k_m + \frac{2\pi k_r}{N} \right) (\mathrm{mod}\, 2\pi) \tag{28}$$

$$= \frac{2\pi k_r}{N}. \tag{29}$$

Since $0 \le k_r < N$, we have that $a \cdot b \in H_N$, and closure is satisfied.

**Associativity** is satisfied if for any $a, b, c \in H_N$, we have $a \cdot (b \cdot c) = (a \cdot b) \cdot c$. For any $a, b, c \in H_N$, we can write $a = {}^{2\pi k_a}/_N$, $b = {}^{2\pi k_b}/_N$, and $c = {}^{2\pi k_c}/_N$ for some integers $k_a$, $k_b$ and $k_c$ in the range $0 \le k_a, k_b, k_c < N$. Then we have

$$(a \cdot b) \cdot c = \left( \left( \frac{2\pi (k_a + k_b)}{N} \right) (\mathrm{mod}\, 2\pi) + \frac{2\pi k_c}{N} \right) (\mathrm{mod}\, 2\pi). \tag{30}$$

Since modulo distributes over addition, the above can be written

$$(a \cdot b) \cdot c = \left( \left( \frac{2\pi k_a}{N} \right) (\mathrm{mod}\, 2\pi) + \left( \frac{2\pi k_b}{N} \right) (\mathrm{mod}\, 2\pi) + \frac{2\pi k_c}{N} \right) (\mathrm{mod}\, 2\pi) \tag{31}$$

$$= \left( \frac{2\pi k_a}{N} \right) (\mathrm{mod}\, 2\pi)(\mathrm{mod}\, 2\pi) + \left( \frac{2\pi k_b}{N} \right) (\mathrm{mod}\, 2\pi)(\mathrm{mod}\, 2\pi) + \left( \frac{2\pi k_c}{N} \right) (\mathrm{mod}\, 2\pi). \tag{32}$$

Using the identity property of modulo (i.e., $(a \bmod b)(\mathrm{mod}\, b) = a \bmod b$), we can write

$$(a \cdot b) \cdot c = \left( \frac{2\pi k_a}{N} \right) (\mathrm{mod}\, 2\pi) + \left( \frac{2\pi k_b}{N} \right) (\mathrm{mod}\, 2\pi)(\mathrm{mod}\, 2\pi) + \left( \frac{2\pi k_c}{N} \right) (\mathrm{mod}\, 2\pi)(\mathrm{mod}\, 2\pi). \tag{33}$$

Using the distributive property of modulo, we can write

$$(a \cdot b) \cdot c = \left( \left( \frac{2\pi k_a}{N} \right) + \left( \frac{2\pi k_b}{N} \right) (\mathrm{mod}\, 2\pi) + \left( \frac{2\pi k_c}{N} \right) (\mathrm{mod}\, 2\pi) \right) (\mathrm{mod}\, 2\pi) \qquad (34)$$

$$= \left( \left( \frac{2\pi k_a}{N} \right) + \left( \frac{2\pi (k_b + k_c)}{N} \right) (\mathrm{mod}\, 2\pi) \right) (\mathrm{mod}\, 2\pi) \qquad (35)$$

$$= a \cdot (b \cdot c), \qquad (36)$$

and associativity is satisfied.

**Identity element.** An element $e \in H_N$ is an identity element if for every $a \in H_N$, we have $e \cdot a = a \cdot e = a$. Consider the element $0 \in H_N$, then for every $a = 2\pi k_a / N \in H_N$ we have

$$a \cdot 0 = \left( \frac{2\pi (k_a + 0)}{N} \right) (\mathrm{mod}\, 2\pi) = a. \qquad (37)$$

By commutativity of addition, we have $0 \cdot a = a$, and $0$ is the identity element of $H_N$.

**Inverse element.** There exists an inverse for every element of the group if for every $a \in H_N$ there exists an element $a^{-1} \in H_N$ so that $a \cdot a^{-1} = a^{-1} \cdot a = e$, where $e \in H_N$ is the identity element. We can write any element $a \in H_N$, in the form $a = 2\pi k_a / N$ where $k_a$ is an integer in the range $0 \le k_a < N$. An inverse element $a^{-1}$ of $a$ satisfying $a \cdot a^{-1} = a^{-1} \cdot a = 0$ can be written in the form $a^{-1} = 2\pi(N - k_a)/N$,

$$a \cdot a^{-1} = \left( \frac{2\pi (k_a + (N - k_a))}{N} \right) (\mathrm{mod}\, 2\pi) = \left( \frac{0}{N} \right) (\mathrm{mod}\, 2\pi) = 0. \qquad (38)$$

By commutativity of addition we have $a^{-1} \cdot a = 0$. It remains to show that $a^{-1}$ is in the set $H_N$. Since $k_a$ is an integer in the range $0 \le k_a < N$, then $N - k_a$ is in the range $0 < N - k_a \le N$, and therefore, $a^{-1} = 2\pi(N - k_a)/N$ is in the set $H_N$ and existence of an inverse is satisfied.

Having shown that the set $H_N$ together with the binary operation $(a \cdot b) \mapsto (a + b) (\mathrm{mod}\, 2\pi)$ satisfies all four axioms of a group, we have that $H_N$ is in fact a group.

## A.2 SATURATION GROUP

We show that the set $S_N$

$$S_N = \left\{ \ldots, -2k/N, -k/N, 0/N, k/N, 2k/N, \ldots \right\}, \qquad (39)$$

generated by $k/N$ with $k$ integer valued, together with the binary operation $\cdot : S_N \times S_N \to S_N$, where

$$a \cdot b \mapsto a + b \qquad (40)$$

for any $a, b \in S_N$ is a group. To do this, we prove the group axioms (18)-(21).

**Closure** is satisfied if for any $a, b \in S_N$, we have $a \cdot b \in S_N$. Any $a, b \in S_N$ can be expressed $a = m_a * k / N$ and $b = m_b * k / N$ for some integers $m_a$ and $m_b$. The product of $a$ and $b$ can then be written

$$a \cdot b = \frac{m_a * k}{N} + \frac{m_b * k}{N} = \frac{(m_a + m_b) * k}{N}. \qquad (41)$$

Since $m = m_a + m_b$ is an integer (the set of integers is closed under addition), we have that $a \cdot b = m * k / N$, is in the set $S_N$, and, closure is satisfied.

**Associativity** is satisfied if for any $a, b, c \in S_N$, we have $a \cdot (b \cdot c) = (a \cdot b) \cdot c$. For any $a, b, c \in S_N$, we can write $a = m_a * k / N$, $b = m_b * k / N$, and $c = m_c * k / N$ for some integers $m_a$, $m_b$, and $m_c$. Then we can write,

$$(a \cdot b) \cdot c = \left( \frac{m_a * k}{N} + \frac{m_b * k}{N} \right) + \frac{m_c * k}{N} = \frac{(m_a + m_b + m_c) * k}{N} \qquad (42)$$

and similarly,

$$c \cdot (b \cdot c) = \frac{m_a * k}{N} + \left( \frac{m_b * k}{N} + \frac{m_c * k}{N} \right) = \frac{(m_a + m_b + m_c) * k}{N}. \qquad (43)$$

Therefore, $a \cdot (b \cdot c) = (a \cdot b) \cdot c$, and associativity is satisfied.

**Identity element.** An element $e \in S_N$ is an identity element if for every $a \in S_N$, we have $e \cdot a = a \cdot e = a$. Consider $0 \in S_N$, then for every $a \in S_N$ where $a = m_a * k / N \in S_N$ we have,

$$a \cdot 0 = \frac{m_a * k}{N} + \frac{0 * k}{N} = a. \tag{44}$$

By commutativity of addition, we have $0 \cdot a = a$, and $0$ is the identity element of $S_N$.

**Inverse element.** There exists an inverse for every element of the group if for every $a \in S_N$ there exists an element $a^{-1} \in S_N$ so that $a \cdot a^{-1} = a^{-1} \cdot a = e$, where $e \in S_N$ is the identity element. We can write any element $a \in S_N$, in the form $a = m_a * k / N$ where $m_a$ is integer valued. An inverse element $a^{-1}$ of $a$ satisfying $a \cdot a^{-1} = a^{-1} \cdot a = 0$ can be written in the form $a^{-1} = -m_a * k / N$,

$$a \cdot a^{-1} = \frac{m_a * k}{N} + \frac{-m_a * k}{N} = 0. \tag{45}$$

By commutativity of addition we have $a^{-1} \cdot a = 0$. It remains to show that $a^{-1}$ is in the set $S_N$. Since $-m_a$ is an integer, $a^{-1} = -m_a * k / N$ is in the set $S_N$ and existence of an inverse is satisfied.

Having shown that the set $S_N$ together with the binary operation $(a \cdot b) \mapsto a + b$ satisfies all four axioms of a group, we have that $S_N$ is in fact a group.

## B   HUE AND SATURATION GROUP ACTION

In this section we prove the proposed hue group action and saturation group action satisfy the axioms for a group action. By definition of a group action, $\varphi$ is a group action if it satisfies the following properties:

1. For all $g, h \in G$ and all $x \in X$

$$\varphi(g, \varphi(h, x)) = \varphi(gh, x) \tag{46}$$

2. For all $x \in X$

$$\varphi(1, x) = x \tag{47}$$

   where $1 \in G$ is the identity element of G.

In the remainder of this section we refer to Equation (46) as 'property one' and Equation (47) as 'property two'.

### B.1   HUE GROUP ACTION

Here we show that the proposed hue group action is indeed a group action. The proposed hue group action on the input space is defined as

$$\varphi_h(h_i, x) = ((x_h + h_i)(\mathrm{mod}\ 2\pi), x_s, x_l), \tag{48}$$

in Equation (3) of the main text.

First we show the proposed hue group action $\varphi_h$ satisfies property one. For any hue shifts $h_i$ and $h_j$, we have

$$\varphi_h(h_i, \varphi_h(h_j, x)) = \varphi_h(h_i, ((x_h + h_j)(\mathrm{mod}\ 2\pi), x_s, x_l)) \tag{49}$$
$$= (((x_h + h_j)(\mathrm{mod}\ 2\pi) + h_i)(\mathrm{mod}\ 2\pi), x_s, x_l)) \tag{50}$$
$$= (((x_h + (h_j + h_i))(\mathrm{mod}\ 2\pi), x_s, x_l)) \tag{51}$$
$$= \varphi_h((h_j + h_i), x), \tag{52}$$

which shows that $\varphi_h$ satisfies the property one. Now we show $\varphi_h$ satisfies property two. We have the identity element hue transformation $h_0 = 0$, and

$$\varphi_h(h_0, x) = ((x_h + h_0)(\mathrm{mod}\ 2\pi), x_s, x_l) \tag{53}$$
$$= (x_h, x_s, x_l) = x, \tag{54}$$

which shows $\varphi_h$ satisfies property two. Having shown that $\varphi_h$ satisfies all properties of a group action, we have that $\varphi_h$ is, in fact, a group action.

## B.2 SATURATION GROUP ACTION

Here we show the proposed saturation group action is a group action. The proposed saturation group action is defined as

$$\varphi_s(s_i, x) = (x_h, \min(x_s + s_i, 255), x_l), \tag{55}$$

in Equation (5) of the main text. First we show the proposed saturation group action $\varphi_s$ satisfies property one. For any saturation shifts $s_i$ and $s_j$, we have

$$\varphi_s(s_i, \varphi_s(s_j, x)) = \varphi_s(s_i, (x_h, \min(x_s + s_j, 255), x_l)) \tag{56}$$
$$= (x_h, \min(\min(x_s + s_j, 255) + s_i, 255), x_l) \tag{57}$$
$$= (x_h, \min(x_s + (s_j + s_i), 255), x_l) \tag{58}$$
$$= \varphi_s((s_j + s_i), x), \tag{59}$$

which shows $\varphi_s$ satisfies property one. Now we show $\varphi_s$ satisfies property two. We have the identity element saturation transformation $s_0 = 0$, and

$$\varphi_s(s_0, x) = (x_h, \min(x_s + s_0, 255), x_l) \tag{60}$$
$$= (x_h, x_s, x_l) = x, \tag{61}$$

which shows $\varphi_s$ satisfies property two. Having shown that $\varphi_s$ satisfies all properties of a group action, we have that $\varphi_s$ is, in fact, a group action.

## C CECONV COMPARISON

In this section, we present qualitative comparisons of our Hue-$N$ equivariant models and the CEConv-$N$ equivariant models proposed in Lengyel et al. (2024).

### C.1 LIFTING LAYER

The lifting layer proposed in CEConv transforms RGB filters in the first layer of the network (see Figure 8b). This approach only outputs valid RGB filters when the hue group is discretized to order N=1 (0 degree rotations) or N=3 (120 degree rotations). For other discretizations of the hue group, CEConv is only approximately equivariant. We detail the CEConv lifting layer in full in Section 4: Lifting layer, and study the impact of invalid RGB rotations in Figure 3.

We propose a lifting layer that transforms input images instead of network filters (see Figure 8a). This seemingly minor change improves equivariance error by more than three orders of magnitude and stabilizes network performance across discretizations.

### C.2 HUE-SHIFT MNIST FEATURE MAP VISUALIZATION

We present visualizations of feature maps produced by the Hue-3, Hue-4, CEConv-3 and CEConv-4 architectures, trained on the hue-shift MNIST dataset. Figure 9 shows a comparison of the feature maps produced by our Hue-3 architecture and the CEConv-3 architecture, and Figure 10 shows a comparison of the feature maps produced by our Hue-4 architecture and the CEConv-4 architecture. The feature map visualizations in Figure 9 are qualitatively similar, but in Figure 10, the CEConv-4 feature map visualization has noticeably more artifacts than the Hue-4 (ours) feature map visualization.

### C.3 COLOR SORTING ON CIFAR-10

We present visualizations of the sorted automobile class using Hue-N and CEConv-N architectures. We trained Hue-3, CEConv-3, Hue-4 and CEConv-4 on CIFAR-10 and used the resulting representations to sort the automobile class by color. Figures 11, 12, 13, and 14 show the automobile class sorted by Hue-3 (ours), CEConv-3, Hue-4 (ours), and CEConv-4 respectively. Figures 11, and 12 are qualitatively similar; however, Figures 13 and 14 are not. In particular, notice that there are blue images in the green band (around the center) of Figure 14, but not in Figure 13.

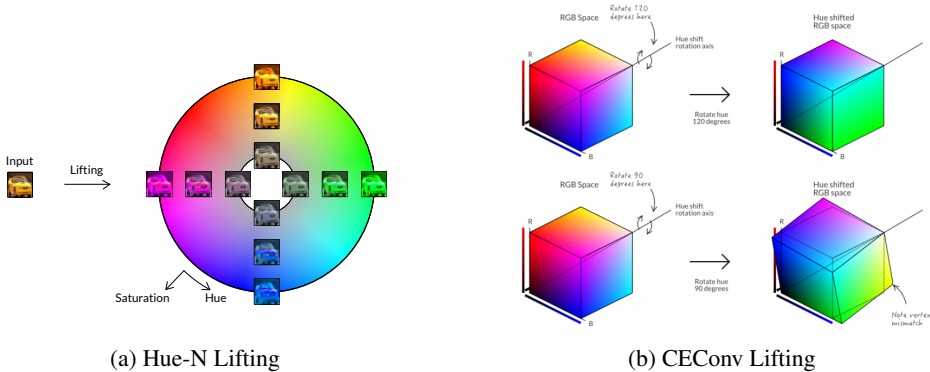

(a) Hue-N Lifting          (b) CEConv Lifting

Figure 8: **Lifting layer. (a)** In Hue-N, an input image (left) is lifted to the hue-saturation group (right) by shifting its hue and saturation values. **(b)** CEConv shifts the hue of a filter in the RGB space by rotating its values about an axis passing through the point $p = (1, 1, 1)$. This approach results in invalid hue rotations for all discretizations of the hue group that are not symmetries of the axis-aligned RGB cube (i.e., $N = 1$ and $N = 3$). (Top) $N = 3$. Rotations symmetric to axis-aligned RGB cube and yields valid rotations. (Bottom) $N = 4$. Rotations yield results where invalid RGB values that lie outside of the RGB cube and needs to be projected.

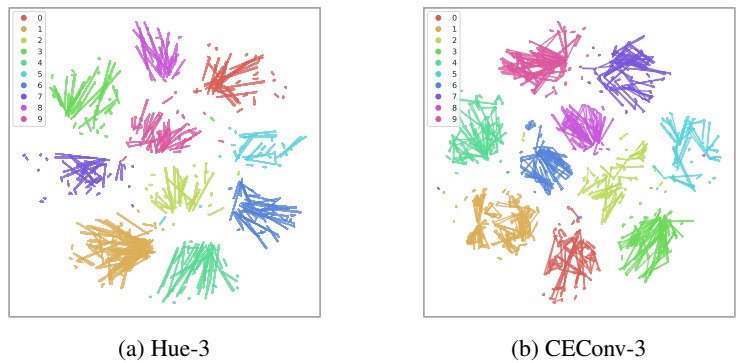

(a) Hue-3          (b) CEConv-3

Figure 9: **Hue shift MNIST feature map visualization.** tSNE projection of hue shifted feature map trajectories in Hue-3 and CEConv-3.

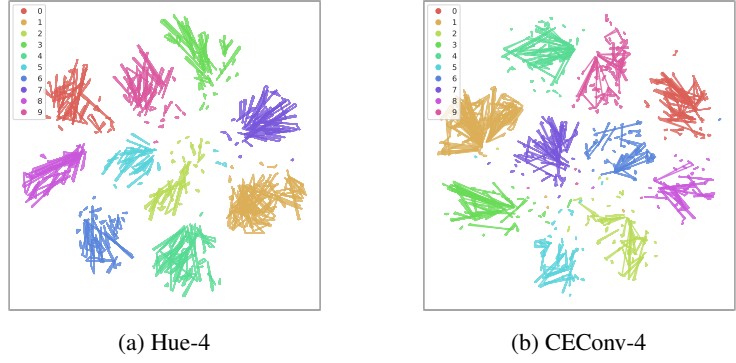

(a) Hue-4          (b) CEConv-4

Figure 10: **Hue shift MNIST feature map visualization.** tSNE projection of hue shifted feature map trajectories in Hue-4 and CEConv-4. In contrast Hue-4, there is more ambiguous scatter in the feature maps of CEConv-4.

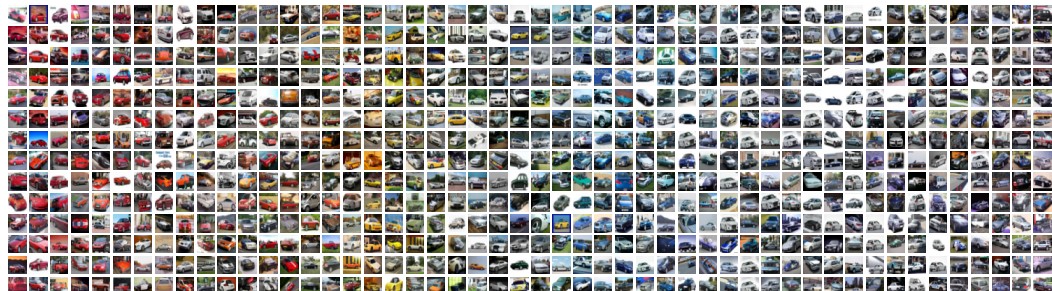

Figure 11: **Color sorting on CIFAR-10 using Hue-3.** We show images from CIFAR-10 automobile class ordered by pairwise distance using Hue-3.

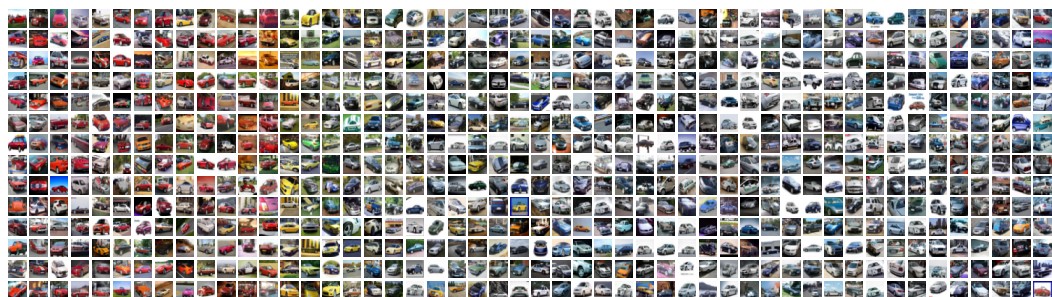

Figure 12: **Color sorting on CIFAR-10 using CEConv-3.** We show images from CIFAR-10 automobile class ordered by pairwise distance using CEConv-3.

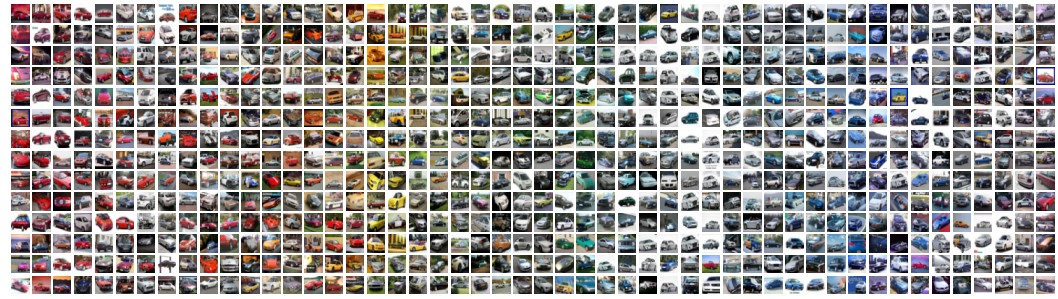

Figure 13: **Color sorting on CIFAR-10 using Hue-4.** We show images from CIFAR-10 automobile class ordered by pairwise distance using Hue-4.

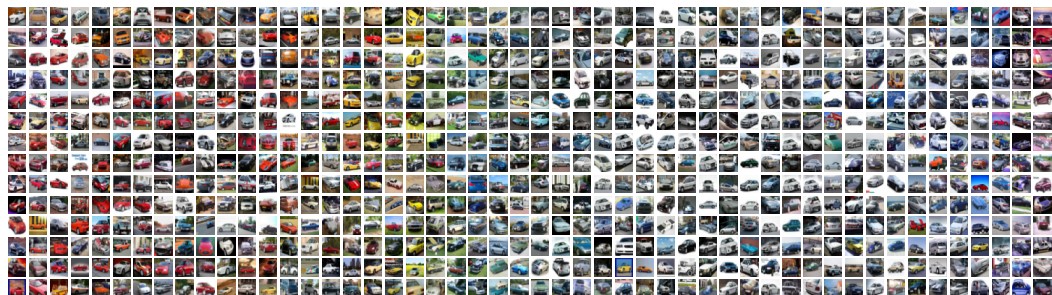

Figure 14: **Color sorting on CIFAR-10 using CEConv-4.** We show images from CIFAR-10 automobile class ordered by pairwise distance using CEConv-4.

## D DATASETS

### D.1 HUE-SHIFT MNIST

We introduce the Hue-shift MNIST dataset to evaluate the performance of our equivariant architectures in the presence of global hue shift. Examples in the training set are randomly assigned a hue between $0°$ and $240°$ (see Figure 15a) and examples in the test set are partitioned into an in-distribution set and an out-of-distribution set. In-distribution examples are randomly assigned a hue from the same distribution as the training set, and out-of-distribution examples are randomly assigned a hue outside of the training set distribution (i.e., between $240°$ and $360°$) (see Figure 15b).

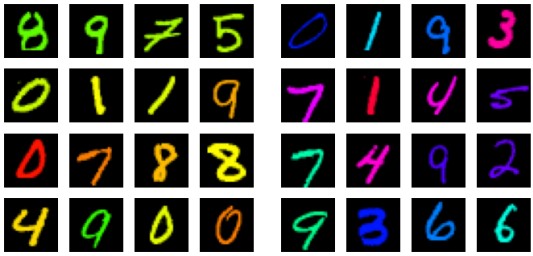

(a) Dataset $A$ examples      (b) Dataset $B$ examples

Figure 15: **Hue-shift MNIST dataset.** (a) Examples from the training dataset and in-distribution testing dataset $A$ are colored with a randomly selected hue between $0°$ and $240°$. (b) Examples from out-of-distribution testing dataset $B$ are colored with a randomly selected hue between $240°$ and $360°$.

### D.2 HUE-SHIFT 3D SHAPES

We introduce the Hue-shift 3D Shapes dataset to evaluate the performance of our equivariant architectures in the presence of local hue shift. Examples in the training set are randomly assigned a hue from the first half of the color space (colors 0-4 as defined in (Burgess & Kim, 2018)) and examples in the test set are are partitioned into an in-distribution set and two out-of-distribution sets. The out-of-distribution test sets are designed to measure robustness to global hue-shift, and local hue-shift. In the out-of-distribution test set designed to measure robustness to global hue-shift, the color of the wall, the floor and the shape are randomly selected from the second half of the color space (colors 5-9 as defined in (Burgess & Kim, 2018)), and in the out-of-distribution test set designed to measure robustness to local hue-shift, the color of the wall and the floor are randomly selected from the first half of the color space, and the color of the shape is randomly selected from the second half of the color space.

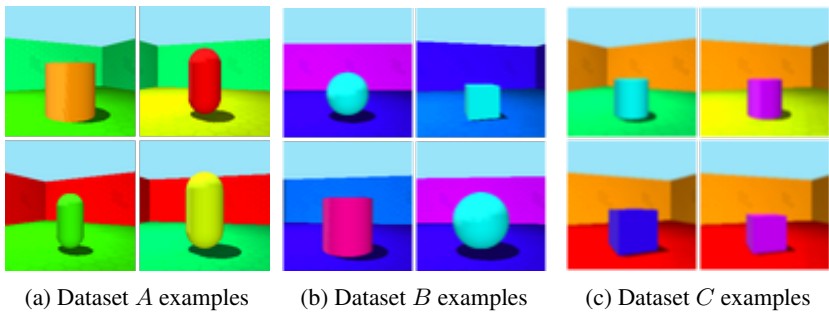

(a) Dataset $A$ examples      (b) Dataset $B$ examples      (c) Dataset $C$ examples

Figure 16: **Hue-shift 3D Shapes dataset.** (a) Examples from the training dataset and in-distribution testing dataset $A$. The color of the wall, the floor and the shape are randomly selected from the first half of the color space (colors 0-4). (b) Examples from the out-of-distribution testing dataset $B$. The color of the wall, the floor and the shape are randomly selected from the second half of the color space (colors 5-9). (c) Examples from the out-of-distribution testing dataset $C$. The color of the wall and the floor are randomly selected from the first half of the color space, and the color of the shape is randomly selected from the second half of the color space.

## D.3 SMALL NORB

We use the small NORB dataset to evaluate the performance of our equivariant architectures in the presence of luminance shifts. Examples in the training set are assigned with medium lighting conditions (2-3 as defined in (LeCun et al., 2004)). and examples in the test set are are partitioned into an in-distribution set and two out-of-distribution sets. Examples in testset $A$ are in-distribution with lighting label 2-3 as defined in LeCun et al. (2004); examples in testset $B$ are out-of-distribution with lower lighting (lighting label 0-1 as defined in LeCun et al. (2004)); and examples in testset $C$ are out-of-distribution with higher lighting (lighting label 4-5 as defined in LeCun et al. (2004)). The out-of-distribution testsets are designed to measure robustness to luminance-shift.

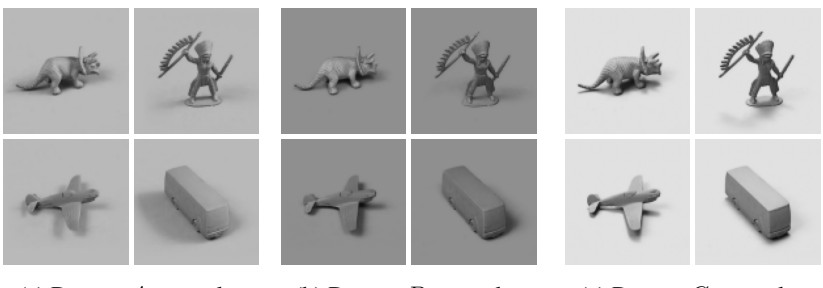

(a) Dataset $A$ examples  (b) Dataset $B$ examples  (c) Dataset $C$ examples

Figure 17: **Small NORB dataset.** (a) Examples from the training dataset and in-distribution testing dataset $A$. (b) Examples from the out-of-distribution testing dataset $B$ with lower lighting conditions. (c) Examples from the out-of-distribution testing dataset $C$ with higher lighting conditions.

## D.4 TINY-IMAGENET

We use the Tiny-ImageNet dataset to evaluate the performance of our equivariant architectures in the presence of color shifts in large datasets. The Tiny-ImageNet dataset (Le & Yang, 2015) is a downsized subset of ImageNet consisting of 100k 64x64 RGB images. Of the 1000 classes represented in ImageNet, 200 are represented in Tiny-ImageNet, each with 500 training examples, 50 validation examples, and 50 unlabeled test examples.

## E  IMPLEMENTATION AND TRAINING DETAILS

In this section we provide architectural details for our equivariant networks, and training details for each experiment. Our source code is publicly available online at `https://github.com/CAB-Lab-Princeton/Learning-Color-Equivariant-Representations`.

All experiments were performed over multiple random seeds to assess the robustness of the model to initialization. Performance statistics on the Hue-shift MNIST, Hue-shift 3D Shapes, and CIFAR-10 datasets were computed over three random seeds (i.e., 1999, 2000, and 2001). Performance statistics the on Camelyon17 dataset were computed over five random seeds (i.e., 1997, 1998, 1999, 2000, and 2001). Performance statistics the on ImageNet dataset were computed over one random seeds (1999).

All models except ImageNet were trained on a shared research computing cluster. Each compute node allocates an Nvidia L40 GPU, 24 core partitions of an Intel Xeon Gold 5320 CPU, and 24GBs of DDR4 3200MHz RDIMMs. ImageNet was trained on compute nodes with 8 Nvidia L40 GPUs, two Intel Xeon Gold 5320 CPUs, and 512GBs of DDR4 3200MHz RDIMMs.

### E.1  HUE-SHIFT MNIST

We compare the classification performance of our architecture and the Z2CNN architecture proposed in Cohen & Welling (2016). Our hue-equivariant architecture has the same number of layers as Z2CNN, but with a reduced filter count to maintain a similar number of parameters at each layer. In the final layer of our hue-equivariant architecture, we perform hue group pooling to yield an hue-invariant representation.

We train our equivariant networks and the conventional architectures for 5 epochs with a batch size of 128. We optimize over a cross-entropy loss using the Adam optimizer (Kingma & Ba, 2014) with

$\beta_1 = 0.9$, and $\beta_2 = 0.999$. We use an initial learning rate of $10^{-3}$ for the $\mathbb{Z}^2$ network, and $10^{-4}$ for the hue-equivariant network. We train CEConv architectures using the hyperparameters reported in (Lengyel et al., 2024).

### E.2 HUE-SHIFT 3D SHAPES

We compare the classification performance of our architecture and the Z2CNN architecture proposed in Cohen & Welling (2016). Our hue-equivariant architectures are designed with the same network structure as the baselines, but with a reduced filter count to maintain a similar number of parameters at each layer. In the final layer of our hue-equivariant architectures, we perform hue group pooling to yield an hue-invariant representation.

Following (Wiles et al., 2022), we train the ResNet architectures for 100k iterations with a batch size of 128. We optimize over a cross-entropy loss using SGD with a learning rate of $10^{-2}$. Images are down-sampled by a factor of 2 to train the CNN architectures. CNN architectures are trained as described in Section E.1.

### E.3 CAMELYON17

We compare the classification performance of our architecture and the ResNet50 architecture (He et al., 2016). Our hue- and saturation-equivariant architectures have the same number of layers as ResNet50, but with a reduced filter count to maintain a similar number of parameters at each layer. In the final layer of our hue- and saturation-equivariant architectures, we perform a group pooling to yield group invariant representation.

We train our equivariant networks and the conventional architectures for 10k iterations with a batch size of 32. We optimize over a cross-entropy loss using the Adam optimizer (Kingma & Ba, 2014) with an initial learning rate of $10^{-2}$, $\beta_1 = 0.9$, and $\beta_2 = 0.999$. We train CEConv architectures using the hyperparameters reported in (Lengyel et al., 2024).

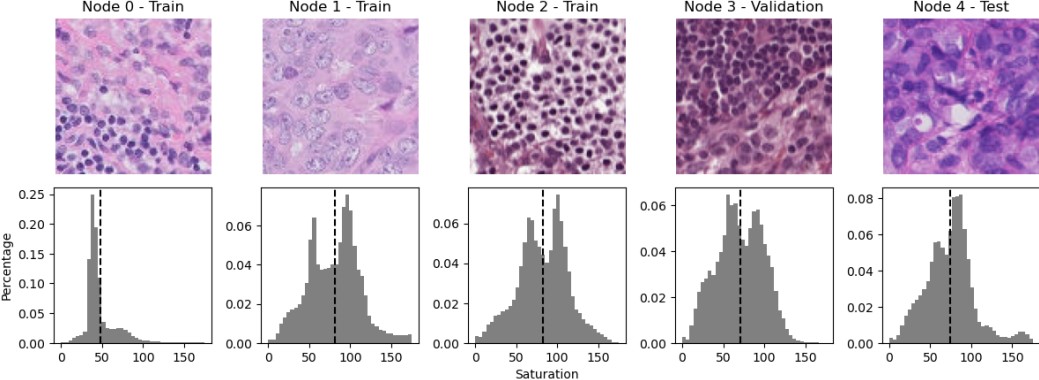

Figure 18: **Camelyon-17 dataset saturation statistics.** Example images and saturation statistics by node (hospital).

We present experimental results for four discretizations of the saturation space. Our choice of discretizations, i.e., $d \in \{1/20, 1/10, 3/20, 1/2\}$, are determined with consideration of the saturation range of the training and validation set (see Figure 18). For a given choice of discretization $d$, an element $s_i \in S_N$ shifts the saturation of an input image by $s_i = i * d * c$, where $c$ is the maximum pixel value. We limit computational expense of lifting and convolution by performing these operations over the truncated set $\{s_{-1}, s_0, s_1\}$ for our Sat-3 model.

### E.4 CIFAR-10

We evaluate the performance of our model in the presence of natural hue shifts on the CIFAR-100 (Krizhevsky et al., 2009) classification dataset. The dataset consists of 60k examples, 50k of which are used for training and 10k for testing.

We compare the classification performance of our architecture and the ResNet44 architecture proposed in Cohen & Welling (2016). Our hue-, saturation- and color-equivariant architectures have the same number of layers as ResNet44, but with a reduced filter count to maintain a similar number of parameters at each layer. In the final layer of our networks, we perform group pooling to yield group invariant representation. We train our equivariant networks and the conventional architectures for 300 epochs, and a batch size of 128. We optimize over a cross-entropy loss using SGD with an initial learning rate of $10^{-1}$ and a cosine-annealing scheduler.

### E.5 CIFAR-100

We evaluate the performance of our model in the presence of natural hue shifts on the CIFAR-100 (Krizhevsky et al., 2009) classification dataset. The dataset consists of 60k examples, 40k of which are used for training, 10k for validation, and 10k for testing.

We compare the classification performance of our architecture and the ResNet44 architecture proposed in Cohen & Welling (2016). Our hue-, saturation- and color-equivariant architectures have the same number of layers as ResNet44, but with a reduced filter count to maintain a similar number of parameters at each layer. In the final layer of our networks, we perform group pooling to yield group invariant representation. We train our equivariant networks and the conventional architectures for 300 epochs, and a batch size of 128. We optimize over a cross-entropy loss using SGD with an initial learning rate of $10^{-1}$ and a cosine-annealing scheduler.

### E.6 CALTECH-101

We optimize over a cross-entropy loss using Adam (Kingma & Ba, 2014) with an initial learning rate of $10^{-2}$. We evaluate the performance of our model in the presence of natural hue shifts on the Caltech-101 (Li et al., 2022) classification dataset. The dataset consists of 9,146 examples, $2/3$ of which are used for training and $1/3$ for testing.

We compare the classification performance of our architecture and the ResNet18 architecture proposed in He et al. (2016). Our hue-, saturation- and color-equivariant architectures have the same number of layers as ResNet18, but with a reduced filter count to maintain a similar number of parameters at each layer. In the final layer of our networks, we perform group pooling to yield group invariant representation. We train our equivariant networks and the conventional architectures for 300 epochs, and a batch size of 16.

### E.7 STL-101

We evaluate the performance of our model in the presence of natural hue shifts on the STL-10 (Coates et al., 2011) classification dataset. The dataset consists of 5k training examples and 8k testing examples.

We compare the classification performance of our architecture and the ResNet18 architecture proposed in He et al. (2016). Our hue-, saturation- and color-equivariant architectures have the same number of layers as ResNet18, but with a reduced filter count to maintain a similar number of parameters at each layer. In the final layer of our networks, we perform group pooling to yield group invariant representation. We train our equivariant networks and the conventional architectures for 300 epochs, and a batch size of 16. We optimize over a cross-entropy loss using Adam (Kingma & Ba, 2014) with an initial learning rate of $10^{-2}$.

### E.8 STANFORD CARS

We evaluate the performance of our model in the presence of natural hue shifts on the Stanford Cars (Krause et al., 2013) classification dataset. As the original host for the Stanford cars dataset is no longer maintained, the dataset was constructed from the images[1], the devkit[2], and the annotation files[3]. The dataset consists of 8,144 training examples and 8,041 testing examples.

---

[1] kaggle.com/datasets/jessicali9530/stanford-cars-dataset
[2] github.com/pytorch/vision/files/11644847/car_devkit.tgz
[3] kaggle.com/code/subhangaupadhaya/pytorch-stanfordcars-classification

We compare the classification performance of our architecture and the ResNet18 architecture proposed in He et al. (2016). Our hue-, saturation- and color-equivariant architectures have the same number of layers as ResNet18, but with a reduced filter count to maintain a similar number of parameters at each layer. In the final layer of our networks, we perform group pooling to yield group invariant representation. We train our equivariant networks and the conventional architectures for 300 epochs, and a batch size of 16. We optimize over a cross-entropy loss using Adam (Kingma & Ba, 2014) with an initial learning rate of $10^{-2}$.

### E.9 OXFORD-IIT PETS

We evaluate the performance of our model in the presence of natural hue shifts on the Oxford-IIT Pets (Parkhi et al.) classification dataset. The dataset consists of 3,680 training examples and 3,669 testing examples.

We compare the classification performance of our architecture and the ResNet18 architecture proposed in He et al. (2016). Our hue-, saturation- and color-equivariant architectures have the same number of layers as ResNet18, but with a reduced filter count to maintain a similar number of parameters at each layer. In the final layer of our networks, we perform group pooling to yield group invariant representation. We train our equivariant networks and the conventional architectures for 300 epochs, and a batch size of 16. We optimize over a cross-entropy loss using Adam (Kingma & Ba, 2014) with an initial learning rate of $10^{-2}$.

### E.10 SMALL NORB

We evaluate the performance of our model in the presence of natural luminance shifts small NORB (Le-Cun et al., 2004) classification dataset. The small NORB dataset consists of 48.6k 96x96 grayscale images captured under 6 different lighting conditions, 9 elevations, and 18 azimuths.

We compare the classification performance of our architecture and the ResNet18 architecture proposed in (He et al., 2016). Our luminance-equivariance architectures have the same number of layers as ResNet18, but with a reduced filter count to maintain a similar number of parameters at each layer. In the final layer of our networks, we perform group pooling to yield group invariant representation. We train our equivariant networks and the conventional architectures for 300 epochs, and a batch size of 16. We optimize over a cross-entropy loss using Adam (Kingma & Ba, 2014) with an initial learning rate of $10^{-2}$.

### E.11 IMAGENET

We evaluate the performance of our model in the presence of natural hue shifts on the ImageNet (Deng et al., 2009) classification dataset. The dataset consists 1000 categories with 14,197,122 annotated images.

We compare the classification performance of our architecture and the ResNet18 architecture proposed in (He et al., 2016). Our hue-equivariant architectures have the same number of layers as ResNet18, but with a reduced filter count to maintain a similar number of parameters at each layer. In the final layer of our networks, we perform group pooling to yield group invariant representation. We train our equivariant networks and the conventional architectures for 100 epochs, and a batch size of 64. We optimize over a cross-entropy loss using Adam (Kingma & Ba, 2014) with an initial learning rate of $10^{-3}$.

## F ADDITIONAL RESULTS

### F.1 LUMINANCE SHIFT SMALL NORB

We demonstrate improved generalization to luminance-shifts compared to ResNet18 (He et al., 2016) on the small NORB classification dataset (LeCun et al., 2004). The small NORB dataset consist of 5 categories of objects under different lighting conditions. Variation in images reflects variation in the location and strength of the lighting. Additional details are provided in Appendix D.3.

**Luminance group and group action.** We expand the notion of color equivariance proposed in (Lengyel et al., 2024) to include equivariance to luminance shifts. In the HSL color space,

luminance can be represented by a real number in the interval $[0, 1]$. Using the approximations presented for construction of the discretized saturation group $S_N$, we similarly identify elements of the discretized luminance group $L_N$, with those of the integers with addition, $(\mathbb{Z}, +)$.

We define the action of the luminance group on HSL images, $x \in X$ where $x : \mathbb{Z}^2 \to \mathbb{R}^3$, and functions on the discrete luminance group, $y \in Y$, where $y : \mathbb{Z}^2 \times L_N \to \mathbb{R}^K$. An element of the luminance group acts on an HSL image by the group action $\varphi_l : L_N \times X \to X$, which shifts the luminance channel of the image. Concretely, for an HSL image $x \in X$ defined as the concatenation of hue, saturation and luminance channels, i.e., $x = (x_h, x_s, x_l)$, the action of an element $l_i$ of the luminance group $L_N$ is given by

$$\varphi_l(l_i, x) = (x_h, x_s, \min(x_l + l_i, c)), \tag{62}$$

where $c$ is the maximum luminance value. An element of the luminance group acts on a function on the discrete luminance group by the group action $\phi_l : L_N \times Y \to Y$, which "translate" the function on the group. Concretely, for a function $f$ on the discrete luminance group $L_N$ defined as the concatenation of functions $f = (f_1, \ldots, f_N)$, the action of an element $l_i$ in the luminance group $L_N$ is given by

$$\phi_l(l_i, f) = (f_{1+i}, \ldots, f_N, \underbrace{\mathbf{0}, \ldots, \mathbf{0}}_{i}). \tag{63}$$

The action of $\varphi_l$ on an input image, and $\phi_l$ on a feature map are shown in Figure 19.

**Generalization to luminance-shift.** We expand the notion of color equivariance proposed in (Lengyel et al., 2024) to capture variation in luminance. The generalization performance of our luminance-equivariant models and Resnet18 (He et al., 2016) are reported in Table 5. Our luminance-equivariant model significantly outperforms the ResNet model in all out-of-distribution cases.

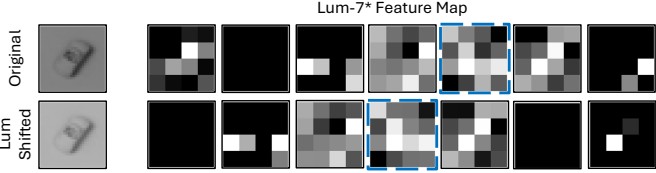

Figure 19: **Luminance equivariant feature maps.** We illustrate the equivariance of our luminance-equivariant model. A luminance shift in the input image space (top-left to bottom-left), results in a feature map translation at each layer of the network (top-right to bottom-right). Corresponding feature maps are highlighted with a blue border.

Table 5: **Generalization to luminance-shift.** Classification error on the small NORB dataset is reported. Our model (Lum-3) achieves improved generalization performance over the conventional CNN model (ResNet18).

| Networks | A/A | A/B | A/C | Params |
|---|---|---|---|---|
| ResNet18 | 8.32 (0.86) | 37.70 (1.51) | 33.88 (5.11) | 11.2M |
| Lum-3* | **7.04 (0.65)** | **18.45 (4.83)** | **25.14 (1.07)** | 11.1M |

## F.2 IMAGENET

We include classification results on the ImageNet dataset (Deng et al., 2009). Additional details are provided in Appendix E.

## F.3 FOREGROUND-BACKGROUND SEGMENTATION

We demonstrate the utility of our method to foreground-background segmentation on the Caltech-101 dataset (Li et al., 2022). The dataset consists of 9,146 examples with annotations containing bounding boxes and contours, which are used to generate ground truth foreground masks. The training set

Table 6: **Generalization to color-shift.** Classification error on the ImageNet dataset is reported. Our model (Hue-3) achieves generalization performance on-par with conventional CNN model (ResNet18) and baseline (CEConv-3).

| Networks | Error | Params |
|----------|-------|--------|
| ResNet18 | 30.66 | 11.7M |
| Hue-3* | 30.28 | 11.4M |
| CEConv-3 | 30.71 | 11.4M |

consists of 6941 examples, the validation set consists of 868 examples, and the test set consists of 868 examples. For uniformity of the dataset, we resize all input images and target masks to have a height and width of 224x224.

We compare the quantitative and qualitative performance of a conventional foreground-background segmentation architecture with our hue-equivariant variant. Each architecture is based on the bottleneck segmentation backbone described in Table 7 where all convolutional layers use residual connections, and each is followed by a batch normalization layer (Ioffe, 2015) and ReLU (Agarap, 2018) activation function. In the final layer, the conventional architecture uses a max pooling layer followed by a sigmoid activation function to produce the predicted foreground-background segmentation mask. Our hue-equivariant architecture is designed similarly, but uses a group pooling layer instead of a max pooling layer.

We report the quantitative performance of our hue-equivariant model and the conventional model in Table 8, and provide a qualitative comparison of the predicted foreground-background segmentation masks in Figure 20.

Table 7: **Foreground-background segmentation network backbone.**

| Layer # | Layer type | Kernel size | Output channels |
|---------|-----------|-------------|-----------------|
| 0 | input | - | 3 |
| 1 | conv | 7x7 | 32 |
| 2 | conv | 7x7 | 64 |
| 3 | conv | 7x7 | 128 |
| 4 | conv | 7x7 | 256 |
| 5 | conv | 1x1 | 256 |
| 6 | conv | 1x1 | 256 |
| 7 | conv | 1x1 | 256 |
| 8 | deconv | 7x7 | 128 |
| 9 | deconv | 7x7 | 64 |
| 10 | deconv | 7x7 | 32 |
| 11 | deconv | 7x7 | 3 |
| 12 | pooling | - | 1 |

Table 8: **Foreground-background segmentation.** Segmentation error on the Caltech-101 dataset is reported. Our proposed color-equivariant convolutions achieve improved performance over the conventional convolutions.

| Convolution | Error | Params |
|-------------|-------|--------|
| Conv | 11.25 (0.12) | 1.1M |
| Hue-3* | **9.67 (0.23)** | 1.2M |

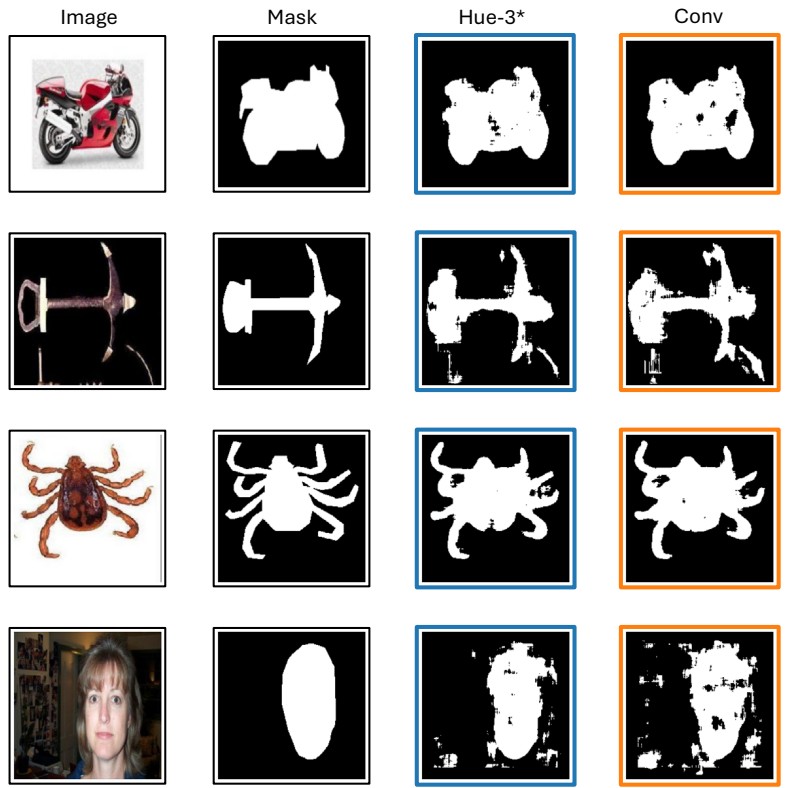

Figure 20: **Foreground-background segmentation.** We show the foreground mask predicted by our approach and a conventional architecture. (Left to right) We show the input image, the ground truth foreground segmentation mask, the foreground segmentation mask predicted by our color-equivariant network, and the foreground segmentation mask predicted by a conventional architecture.

