# OpenReview forum: "Learning Color Equivariant Representations"
_ICLR.cc/2025/Conference — ICLR 2025 Poster_

### Official Review · Reviewer_r9ZC · 2024-10-30

**Soundness:** 3
**Presentation:** 3
**Contribution:** 2
**Rating:** 3
**Confidence:** 3

**Summary:**

The authors propose GCNNs which are equivariant to color variation. GCNNs have been widely used for geometric transformations while only a few research has attempted to apply GCNNs to color-equivariant networks to achieve generalization to out-of-distribution samples. Throughout the paper, the authors reasonably justified the reason for designing the color-equivariant networks and proposed a proper method for dealing with such situations. Both quantitative and qualitative evaluations are conducted in the paper.

**Strengths:**

- Interpretable visual illustrations of the results
- Interesting motivation
- Decent paper writing

**Weaknesses:**

Overall, it is clear to me that the proposed method is better than the baselines (ResNet50, Z2CNN) under the generalization testing setup. However, the improvements over CEConvs look marginal (performance-wise and method-wise). Other weaknesses I have found are:

1. Related Works:
    - What is the connection/difference of the proposed setting with Domain Generalization?
2. Missing baselines to support the argument in L44-50.
    - converting input image to grayscale
    - data augmentation

**Questions:**

1. Fig. 1: Baseline comparison with CEConvs

2. Fig. 4
    - What is the “Model sample efficiency”?
    - How can the error improvement be computed?
    - Fig. 4 shows that there is no improvement over Z2CNN if 10% or 100% of training example is used. Why does it happen?
    - On the other hand, the advantage of the proposed methods are shown effective when only a little portion of the training example is used. What does this mean?

3. Table 1: Baseline comparison with CEConvs

4. Fig 5: Baseline comparison with CEConvs

5. Fig 7: Baseline comparison with CEConvs

6. What is the difference between the global hue-shift out-of-distribution (A/B) and the local hue-shift out-of-distribution (A/C)?

---

> ### Author Response · Authors · 2024-11-21
>
> > What is the connection/difference of the proposed setting with Domain Generalization?
>
> The goal of domain generalization is to learn representations that are invariant to distribution shift [1,2,3]. This is in contrast to the related but distinct goal of equivariant representation learning which is to learn representations that are equivariant to (i.e., transform predictably in response to) specific transformations of the inputs. While not specifically designed for generalization, equivariant neural networks have been shown to improve model generalization performance (both empirically and theoretically [4,5,6,7]) when the specific transformations are symmetries (or nuisance transformations) of the underlying task. We provide mathematical definitions of invariance and equivariance in Section 3: Equivariance.
>
> [1] Gulrajani, Ishaan, and David Lopez-Paz. "In search of lost domain generalization." arXiv preprint arXiv:2007.01434 (2020).
> [2] Blanchard, Gilles, Gyemin Lee, and Clayton Scott. "Generalizing from several related classification tasks to a new unlabeled sample." Advances in neural information processing systems 24 (2011).
> [3] Muandet, Krikamol, David Balduzzi, and Bernhard Schölkopf. "Domain generalization via invariant feature representation." International conference on machine learning. PMLR, 2013.
> [4] Behboodi, Arash, Gabriele Cesa, and Taco S. Cohen. "A pac-bayesian generalization bound for equivariant networks." Advances in Neural Information Processing Systems 35 (2022): 5654-5668.
> [5] Elesedy, Bryn. "Group symmetry in pac learning." ICLR 2022 workshop on geometrical and topological representation learning. 2022.
> [6] Zhu, Sicheng, Bang An, and Furong Huang. "Understanding the generalization benefit of model invariance from a data perspective." Advances in Neural Information Processing Systems 34 (2021): 4328-4341.
> [7] Brehmer, Johann, et al. "Does equivariance matter at scale?." arXiv preprint arXiv:2410.23179 (2024).
>
> > Missing baselines to support the argument in L44-50. converting input image to grayscale  data augmentation
>
> We ran additional experiments to compare the performance of our color equivariant approach to baselines that use data augmentation or grayscale input images and found that our approach performed on-par or better than these color invariant approaches. We have included these results in Table 4. We’d like to point out that even though all methods perform similarly on classification, our color equivariant architecture is more interpretable by construction, has higher sample efficiency (which others have shown both theoretically and empirically [4,5,6,7], and we’ve shown Figure 4), and naturally allows for downstream tasks such as color based sorting (see Figure 7) and color alignment.
>
> > What is the “Model sample efficiency”? The advantages of the proposed methods are evident only when a small proportion of the training example is used. What does this mean?
>
> We use the definition of sample efficiency given in [8]: “Sample efficiency refers to the ability of a statistical method or algorithm to make accurate inferences or predictions using a minimum amount of sample data.” In Figure 4, we empirically show that our models have greater sample efficiency than conventional CNNs by plotting the performance of each model when trained on 100%, 10%, 1%, and 0.1% percent of the examples in the training dataset.
>
> Our models outperform the conventional CNN when less than 10% of the training examples are used. This happens because our network architecture is constrained to be equivariant to hue variation. By constraining our network we achieve strong generalization performance even when only a small number of samples are used (i.e., high sample efficiency). The generalizability of representations learned in equivariant neural networks has been observed empirically and proved theoretically in [4,5,6,7].
>
> [8] “Sample Efficiency - (Sampling Surveys) - Vocab, Definition, Explanations | Fiveable.” Fiveable.me, 2024, fiveable.me/key-terms/sampling-surveys/sample-efficiency. Accessed 20 Nov. 2024.
>
> > How can the error improvement be computed?
>
> We compute the error improvement as the difference between the classification error of Hue-N (ours) and the classification error of the conventional CNN. We chose to show the error improvement rather than the absolute error for clarity of the visual presentation.

---

> > ### Author Response · Authors · 2024-11-21
> >
> > > Include comparisons to CEConv in Figure 1, Table 1, Figure 5 and Figure 7
> >
> > **Figure 1** illustrates our proposed method. We show the equivariance of our feature representations on the left, and a depiction of our proposed lifting layer on the right. A visualization of CEConv feature maps transforming under hue shift is shown in Figure 6, and an illustration of the CEConv lifting layer is shown in Appendix B.1. In contrast to our approach, CEConv is only equivariant when the transformation group is discretized to order N=1 or N=3; for other discretizations CEConv is only approximately equivariant. This effect is a consequence of their choice of lifting layer which transforms the RGB cube, producing invalid values for all rotations except 0 degrees (i.e., N=1) and 120 degrees (i.e., N=3). We study this effect and show results in Figure 3, and describe their lifting layer in detail in Section 4: Lifting layer.
> >
> > **Regarding Table 1**. We added the performance of CEConv-3 and CEConv-4 on hue-shift MNIST in Table 1. The performance of CEConv-3 is comparable to our Hue-$N$ networks, but the performance of CEConv-4 is significantly worse. We attribute the poorer performance of CEConv-4 to the CEConv lifting layer which produces invalid RGB values for all hue rotations except 0 degrees (i.e., N=1) and 120 degrees (i.e., N=3). We describe the CECov lifting layer in detail in Section 4: Lifting layer, and study the impact of invalid RGB values in Figure 3.
> >
> > **Regarding Figure 5**. We added feature map visualizations of CEConv-3 and CEConv-4 networks trained on hue-shift MNIST in Appendix B.2. Figure 9 shows a comparison of the feature maps produced by our Hue-3 (ours) architecture and the CEConv-3 architecture, and Figure 10 shows a comparison of the feature maps produced by our Hue-4 (ours) architecture and the CEConv-4 architecture. The feature maps visualizations in Figure 9 are qualitatively similar, but in Figure 10, the CEConv-4 feature map visualization has noticeably more artifacts than the Hue-4 (ours) feature map visualization.
> >
> > **Regarding Figure 7**. We added visualizations of the sorted automobile class using CEConv in Appendix B.3. We trained CEConv-3 and CEConv-4 on CIFAR-10 and used the resulting representations to sort the automobile class by color. Figures 11, 12, 13, and 14 show the automobile class sorted by Hue-3 (ours), CEConv-3, Hue-4 (ours), and CEConv-4 respectively. Figures 11 and 12 are qualitatively similar; however, Figures 13 and 14 are not. In particular, notice that there are blue images in the green band (around the center) of Figure 14, but not in Figure 13.
> >
> > > What is the difference between global hue-shift out-of-distribution (A/B) and the local hue-shift out-of-distribution (A/C)?
> >
> > In the global hue-shift out-of-distribution experiment (A/B), all images in the training set are comprised of components (i.e., the wall, the floor and the shape) whose color is drawn from distribution A (i.e., colors 0-4 as defined in [9]). In the test set, all images are comprised of components whose color is drawn from distribution B (i.e., colors 5-9 as defined in [9]). We call this experiment global hue-shift since the color of all components in a training set image are drawn from one distribution and the color of all components in a test set image are drawn from another distribution.
> >
> > In the local hue-shift out-of-distribution experiment (A/C), all images in the training set are comprised of components whose color is drawn from distribution A. In the test set, the color of some components in an image are drawn from distribution A and the color of other components are drawn from distribution B. Specifically, the color of the wall and the floor are drawn from distribution A, and the color of the shape is drawn from distribution B. We call this experiment local hue-shift since not all components of an image in the test set are drawn from a different distribution than the components of an image in the training set.
> >
> > We describe the hue-shift 3D shapes dataset in more detail in Appendix C.2.
> >
> > [9] Chris Burgess and Hyunjik Kim. 3D Shapes dataset. https://github.com/deepmind/3dshapes-dataset/, 2018.

---

> ### Comment · Reviewer_r9ZC · 2024-11-30
>
> I appreciate the authors' efforts for their responses to my questions.
> However, I still have remaining concerns as written below. Thus, I would keep my initial rating.
>
> 1. Table 1 results:
>   - The performance improvement of Hue-3 over CEConv-3 is marginal.
>   - The fact that CEConv-4 is worse is not meaningless, but it cannot be an evidence that the proposed method is better than the baseline because of the marginal gap between Hue-3 over CEConv-3.
>
> 2. Table 2 results:
>   - The performance improvements of Hue-3 and Hue-4 over CEConv-3 are marginal.
>
> 3. Table 4 results:
>   - It is not intuitively understandable how ResNet-gray and ResNet-aug could be worse than ResNet for color-shift generalization performance. Specific descriptions such as experiment settings and justifications are needed.
>   - If we compare Hue-4 with ResNet or ResNet-aug, the proposed method show worse performance in 3 columns (Cifar-10, Cifar-100, and STL-10) out of 6 columns. To me, the results from Table 4 are not good enough to support the argument that the proposed methods are more robust to color-shift.
>   - Additional benefits of the proposed method over ResNet-aug and ResNet-gray (more interpretable, having higher sample efficiency, naturally allowing for downstream tasks) are not persuasive enough yet. — Why is it important in practice achieving the additional benefits? For example, as for the “model sample efficiency”, the proposed methods show better performance when the # of training samples are less than 5% of the whole dataset. This setting sounds somehow unrealistic to me.
>
> 4. Question: what is Model sample efficiency of the baseline (CEConv)? — I am wondering if Fig. 4 is the only benefit of the proposed methods or the benefit that CEConv also has.

---

> > ### Author Response · Authors · 2024-12-02
> >
> > Thank you for engaging with our rebuttal. We’d like to clarify a few points below.
> >
> > > The performance improvement of Hue-3 over CEConv-3 is marginal. .. CEConv-4 is worse.
> >
> > Exactly! The performance of CEConv deteriorates when $N$ is not equal to three or one, whereas our performance remains stable. This poses a problem in practice since complementary colors (i.e. colors that are 180 degrees apart) are arguably more prevalent in man made spaces, which suggests that even valued $N$ are more useful than odd valued $N$. Additionally, there are settings (e.g., fine-grained classification) where it may be preferable to select a large value for $N$ to better approximate the continuous hue group. In effect, the method proposed in CEConv requires practitioners to make a choice between practicality and invertibility error, while ours does not.
> >
> > > Specific descriptions of ResNet-gray and ResNet-aug experiment settings and justifications are needed.
> >
> > In Table 4 we note that ResNet-aug and Resnet-gray denote a ResNet architecture trained on a color-jitter and grayscale image dataset respectively. For each experiment, we give details of the specific ResNet architecture in Appendix D. The color-jitter dataset is generated by randomly shifting the hue channel of an input image by a hue between -180 and 180 degrees. The grayscale image dataset is generated by converting each input image to grayscale. Thank you for pointing this out, we will add the details for color-jitter and grayscale to our paper.
> >
> > > Why is it important in practice achieving the benefits of interpretability and higher sample efficiency?
> >
> > Interpretability of learned solutions is important in domains as diverse as explainable AI, safe AI and scientific discovery. A model with higher sample efficiency can learn more effectively from smaller datasets. This is valuable in settings such as robotics, and medical imaging where the cost of data acquisition is high.
> >
> > > What is Model sample efficiency of the baseline (CEConv)?
> >
> > The higher sample efficiency of our approach is a consequence of constraining our architecture to be equivariant to hue variation. We would expect to see this benefit with CEConv as well, but perhaps to a lesser degree due to the observed equivariance error. We will explore this and include the results in our paper.

---

### Official Review · Reviewer_7Wsa · 2024-10-31

**Soundness:** 4
**Presentation:** 3
**Contribution:** 2
**Rating:** 5
**Confidence:** 3

**Summary:**

This paper builds on the approach “Color Equivariant Convolutional Networks” [16, Neurips 2023], modifying their lifting layer by operating on the input image rather than on the filters. Additionally, it generalizes the equivariance, previously only defined on the hue channel to the  saturation channel. Experiments on synthetic datasets (hue shift MNIST and hue shift 3D shape) and on CIFAR, Caltech, Camelyon17, Caltech, STL-101, Stanford Cars, and Oxford pets dataset show improved classification performance compared to the CE-conv3 approach [16] and to Resnet baselines with same number of parameters.

**Strengths:**

* The paper is well written and illustrated.
* The approach is well motivated and addresses concrete tasks, for instance improving classification robustness in case of different imaging processes coming from different labs in the case of medical imagery.
* The claims are supported by experiments.
* Some limitations of the approach are presented.
* The code will be open source.
* The appendix provides necessary information to reproduce results.
* The approach designs color equivariance in group CCNN models, which is an interesting property.

**Weaknesses:**

* The claim of improved equivariance by order of magnitude over [16] on the 3D shape dataset is true but the error of [16] is already very low on this dataset, 0.05 at most for CEconv-3. As the error is very small, this difference in magnitude will not translate into drastic improvements in practice.
* [Comparison to stronger baselines] The comparison to [16] on non-synthetic datasets displays larger errors than in [16]. Why not use their experimental protocol that seems well detailed in their paper?
[Incomplete comparison] Only results with CE-conv3 and CE-conv4 appear in the current paper, while [16] often shows better results with CE-conv2.
* [No comparison to SOTA] Further discussion on the Camelyon-17 results would be interesting: Looking at the leaderboard on this dataset, https://wilds.stanford.edu/leaderboard/ , we note a number of approaches are performing better than the proposed approach. What would be the pros and cons of using the proposed approach vs the SOTA ?
* The cardinality N of the group used in the experiments is not discussed. In addition, it is only introduced on page 7, it would help the reader to define the notation before as all figures are using it.

**Questions:**

1/ Why not use the experimental protocol of [16] that seems well detailed in their paper for Table 4?

2/ What would be the pros and cons of using the proposed approach vs the SOTA (or a strong baseline from the leaderboard such as Group DRO) on Camelyon17 ?

3/ Could you explain how the different cardinalities have been chosen for the different experiments?

4/ In the limitation section, the computation overhead of the approach is mentioned, without mentioning explicit numbers compared to the baseline’s training time, could this information be mentioned?

---

> ### Author Response · Authors · 2024-11-21
>
> > The error of CEConv-3 is already very low on 3D shapes.
>
> The out-of-distribution error of CEConv-3 is small on the 3D shapes dataset, but the out-of-distribution error of CEConv-4 is significantly worse. This is because hue rotations of the RGB cube are only valid when the cardinality of the discretized hue group is one or three. In Figure 3, we show that the cardinality $N$ of the group (three for CEConv-3 and four for CEConv-4) impacts the invertibility of hue rotations. The invertibility error is negligible when N=1 and N=3, but greater than 7% otherwise. This is important since the equivariance error is determined by the invertibility error (it is only low when N=1 and N=3). It poses a problem in practice since complementary colors (i.e. colors that are 180 degrees apart) are arguably more prevalent in man made spaces, which suggests that even valued $N$ are more useful than odd valued $N$. Additionally, there are settings (i.e., fine-grained classification) where it may be preferable to select a large value for $N$ to better approximate the continuous hue group. In effect, the method proposed in [16] requires practitioners to make a choice between practicality and invertibility error, while ours does not.
>
> > Why not use their experimental protocol detailed in [16]?
>
> The experimental protocol described in [16] does not provide sufficient details for reproduction. The batch sizes, learning rate scheduler parameters, and weight decay are not provided for any experiments. Additionally, train/test splits are not provided for the Caltech-101 dataset. We attempted to reproduce their results but ultimately decided to use the same hyperparameters for all networks (we provide all hyperparameters necessary for reproduction in Appendix D), and use a train-test split of 67%-33% for Caltech-101.
>
> > Only results with CE-conv3 and CE-conv4 appear in the current paper, while [16] often shows better results with CE-conv2.
>
> Thank you for pointing out the notational conflict between our paper and [16]. In our paper, the $N$ in CEConv-N refers to the choice of discretization for the hue group so that a higher value of $N$ gives a better approximation to the continuous hue group. In [16], it refers to the number of layers in the network that are constrained to be color equivariant. This difference is significant and has several implications. First, their model is only equivariant when all ResNet stages are constrained to be equivariant (i.e., N=3 for CIFAR, N=4 otherwise), whereas all the networks we introduce are fully equivariant. Moreover, all of their experiments are performed with the same discretization of the hue group, i.e., cardinality three. Our experiments and analyses show that the performance of [16] deteriorates when $N$ is not equal to three, whereas our performance remains stable. We provide a comparison below to summarize the notational conflict and will use a different notational convention in the camera ready version.
>
> * **Ours**: CEConv-N, N denotes the cardinality of the hue transformation group
> * **[16]**: CEConv-N, N denotes the number of ResNet stages that are constrained to be color equivariant.

---

> > ### Author Response · Authors · 2024-11-21
> >
> > > What would be the pros and cons of using the proposed approach vs the SOTA (or a strong baseline from the leaderboard such as Group DRO) on Camelyon17?
> >
> > Many of the top leaderboard methods for Camelyon17 (without unlabeled data) are domain generalization methods, whereas our approach is an equivariant representation learning method. These two approaches differ in their goal and strengths. The goal of domain generalization methods is to learn representations that are invariant to distribution shift [1,2,3]; and the goal of equivariant representation learning methods is to learn representations that are **equivariant** to (i.e., transform predictably in response to) specific transformations of the inputs.
> >
> > Some pros and cons of using our proposed approach vs strong baselines from the leaderboard are the following:
> > * **Con #1**: Color equivariant approaches can be more challenging to design than color invariant approaches, since they require identification of the color geometry with a geometric transformation group and implementation of a GCNN.
> > * **Pro #1**: Color equivariant approaches are more interpretable by construction.
> > * **Pro #2**: Color equivariant approaches have higher sample efficiency (see Figure 4 in our paper, and [4,5,6,7] who highlight the sample efficiency of equivariant representation learning both empirically and theoretically).
> > * **Pro #3**: Color equivariant approaches naturally allow for downstream tasks such as color based sorting (see Figure 7) and color alignment.
> >
> > [1] Gulrajani, Ishaan, and David Lopez-Paz. "In search of lost domain generalization." arXiv preprint arXiv:2007.01434 (2020).
> > [2] Blanchard, Gilles, Gyemin Lee, and Clayton Scott. "Generalizing from several related classification tasks to a new unlabeled sample." Advances in neural information processing systems 24 (2011).
> > [3] Muandet, Krikamol, David Balduzzi, and Bernhard Schölkopf. "Domain generalization via invariant feature representation." International conference on machine learning. PMLR, 2013.
> > [4] Behboodi, Arash, Gabriele Cesa, and Taco S. Cohen. "A pac-bayesian generalization bound for equivariant networks." Advances in Neural Information Processing Systems 35 (2022): 5654-5668.
> > [5] Elesedy, Bryn. "Group symmetry in pac learning." ICLR 2022 workshop on geometrical and topological representation learning. 2022.
> > [6] Zhu, Sicheng, Bang An, and Furong Huang. "Understanding the generalization benefit of model invariance from a data perspective." Advances in Neural Information Processing Systems 34 (2021): 4328-4341.
> > [7] Brehmer, Johann, et al. "Does equivariance matter at scale?." arXiv preprint arXiv:2410.23179 (2024).
> >
> > > Cardinality is not defined until after it has been used.
> >
> > Thank you for pointing this out! We’ve added a definition of cardinality in the method section.
> >
> > > Could you explain how the different cardinalities have been chosen for the different experiments?
> >
> > We selected cardinalities of $N=3$ and $N=4$ for our experiments. We performed experiments using $N=3$ for a fair comparison against previous work ([16]). We performed experiments using $N=4$ to show that our network is robust to the choice of hue group discretization, while [16] is not. This limitation of [16] is discussed in Section 4: Lifting layer, and Figure 3.
> >
> > > Can the computation overhead of the approach be mentioned?
> >
> > Yes, we now mention the computational overhead of the approach in our conclusion. As described in [8], the computational cost of GCNNs is approximately equal to that of conventional CNNs with a filter bank size equal to that of the augmented filter bank used in GCNNs.
> >
> > [8] Cohen, Taco, and Max Welling. "Group equivariant convolutional networks." International conference on machine learning. PMLR, 2016.

---

> > > ### Comment · Reviewer_7Wsa · 2024-11-27
> > > **The practical interest of the approach remains to be shown**
> > >
> > > I would like to thank the authors for their discussion on the comparison of their approach on the Camelyon dataset. I understood that their results are not comparable to the one of the leaderboard because addressing a different problem. However, to me, the main positive aspect of this work is its motivation provided in l. 40 to 42: "Consider, for instance, the case of medical imaging where images of tissue samples are collected from different labs..." exactly describing the distribution shift problem.
> > > The authors cited 3 advantages of their approach (interpretability, sample efficiency, efficiency in color sorting tasks).
> > > I agree on interpretability and color sorting tasks, but the sample efficiency is only shown on MNIST. There is no evidence in this work that the proposed approach would be useful in a real world task.
> > >
> > > In their answer to my comment:
> > > > The claim of improved equivariance by order of magnitude over [16] on the 3D shape dataset is true but the error of [16] is already very low on this dataset, 0.05 at most for CEconv-3. As the error is very small, this difference in magnitude will not translate into drastic improvements in practice.
> > > the authors wrote:
> > > >  Additionally, there are settings (i.e., fine-grained classification) where it may be preferable to select a large value for
> > >  to better approximate the continuous hue group. In effect, the method proposed in [16] requires practitioners to make a choice between practicality and invertibility error, while ours does not.
> > >
> > > Why not showing the positive aspect of this work on fine grained classification tasks?

---

> > > > ### Author Response · Authors · 2024-11-30
> > > >
> > > > Thank you for engaging with our rebuttal. There are a couple of points we’d like to clarify.
> > > >
> > > > >Citing the three orders of magnitude improvement is misleading because classification accuracy is about the same.
> > > >
> > > > While the improved equivariance error does not lead to notable improvements in classification error on hue-shift 3D shapes, it does on hue-shift MNIST (see Table 1). Moreover, we can see the qualitative impact of improved equivariance error on the CIFAR-10 automobile sorting task (see Appendix B.3). Figures 11, 12, 13, and 14 show the automobile class sorted by Hue-3 (ours), CEConv-3, Hue-4 (ours), and CEConv-4 respectively. Figures 11 and 12 are qualitatively similar; however, Figures 13 and 14 are not. In particular, notice that there are blue images in the green band (around the center) of Figure 14, but not in Figure 13.
> > > >
> > > > > Sample efficiency is only shown on MNIST.
> > > >
> > > > While our sample efficiency experiments are restricted to the hue-shift MNIST dataset, we expect our results will generalize to other datasets. Our expectation is grounded in results from previous works [1,2,3,4] which show, both theoretically and empirically, that equivariant neural networks (such as ours) achieve better sample efficiency. The intuition for the theoretical argument is that with a neural network equivariant to a specific transformation group G, transformations g x of a sample x (where g is an element of G) do not need to be explicitly learned (e.g., by data augmentation) as they would in a conventional architecture, so the learning space is reduced and the sample efficiency is increased.
> > > >
> > > > [1] Behboodi, Arash, Gabriele Cesa, and Taco S. Cohen. "A pac-bayesian generalization bound for equivariant networks." Advances in Neural Information Processing Systems 35 (2022): 5654-5668.
> > > > [2] Elesedy, Bryn. "Group symmetry in pac learning." ICLR 2022 workshop on geometrical and topological representation learning. 2022.
> > > > [3] Zhu, Sicheng, Bang An, and Furong Huang. "Understanding the generalization benefit of model invariance from a data perspective." Advances in Neural Information Processing Systems 34 (2021): 4328-4341.
> > > > [4] Brehmer, Johann, et al. "Does equivariance matter at scale?." arXiv preprint arXiv:2410.23179 (2024).
> > > >
> > > > > There is no evidence in this work that the proposed approach would be useful in a real world task.
> > > >
> > > > We agree that applicability to real world tasks is important and have presented results across multiple real world datasets including tumor classification on Camelyon17 [5] (see Table 3), fine-grained classification on Stanford Cars [6] and Oxford Pets [7] (see Table 4), and foreground-background segmentation on Caltech-101 [8] (see Appendix E.3). In each of these experiments, our equivariant models outperform reasonable baselines while providing the added benefit of improved interpretability and sample efficiency by construction.
> > > >
> > > > [5] Litjens, Geert, et al. "1399 H&E-stained sentinel lymph node sections of breast cancer patients: the CAMELYON dataset." GigaScience 7.6 (2018): giy065.
> > > > [6] Krause, Jonathan, et al. "3d object representations for fine-grained categorization." Proceedings of the IEEE international conference on computer vision workshops. 2013.
> > > > [7] Parkhi, Omkar M., et al. "Cats and dogs." 2012 IEEE conference on computer vision and pattern recognition. IEEE, 2012.
> > > > [8] Li, F.-F., Andreeto, M., Ranzato, M., & Perona, P. (2022). Caltech 101 (1.0) [Data set]. CaltechDATA. https://doi.org/10.22002/D1.20086
> > > >
> > > > > Why not showing the positive aspect of this work on fine grained classification tasks?
> > > >
> > > > Actually, we do! In Table 4, we note the performance of our model against several baselines on the Stanford Cars [6] and Oxford Pets[7] fine-grained classification datasets. In each case, our hue-equivariant models outperform all baseline models. Our model outperforms competitive baselines by 2.00% on Stanford Cars and 2.67% on Oxford Pets. Thank you for your suggestion, we will highlight these results in the camera ready version.

---

> > ### Comment · Reviewer_7Wsa · 2024-11-27
> >
> > Thank you for your answers, and sorry for responding only now.
> >
> > > the out-of-distribution error of CEConv-4 is significantly worse
> >
> > The error of CEConv-4 is 0.6 ? Yes it is significantly worse than 0.02 but still for a classification accuracy error, it means that the shapes are correctly classified 99.4 % of the time?  If that is the case, I still think citing the three orders of magnitude improvement, in the paper and in the abstract is misleading because in practice, the classification accuracy is about the same.
> >
> > > The experimental protocol described in [16] does not provide sufficient details for reproduction.
> >
> > > We provide a comparison below to summarize the notational conflict and will use a different notational convention in the camera ready version.
> >
> > OK, thanks for these clarifications.

---

### Official Review · Reviewer_WTM5 · 2024-11-01

**Soundness:** 3
**Presentation:** 3
**Contribution:** 2
**Rating:** 6
**Confidence:** 3

**Summary:**

The paper presents a novel convolutional neural network (CNN) architecture tailored to handle color variations effectively. The authors introduce a color-equivariant group convolutional network (GCNN) that is robust to changes in hue and saturation, significantly improving performance in tasks involving perceptual color transformations. By employing a new lifting layer that operates directly on input images rather than filters, the model overcomes limitations of existing color-equivariant networks, achieving enhanced stability and reducing equivariance errors by orders of magnitude. This approach yields superior performance on both synthetic and real-world datasets, particularly excelling in generalizing to out-of-distribution color variations and enhancing sample efficiency.

**Strengths:**

1, The paper presents a novel approach to color-equivariant convolutional neural networks (CNNs) by extending GCNNs to handle hue and saturation changes in the HSL color space. This is achieved with an inventive lifting layer that acts directly on input images, improving stability and reducing errors by avoiding invalid RGB values—a common issue in previous architectures.

2, The authors demonstrate a thorough experimental design using various synthetic and real-world datasets (e.g., Hue-shift MNIST, Camelyon17) known for color variations, showing the model’s robustness. Both quantitative and qualitative analyses, including sample efficiency and feature map visualizations, offer a detailed evaluation. Comparisons with competitive baselines like CEConv and ResNet confirm the model’s generalizability and efficiency.

3, The paper is well-structured, with a logical flow from problem statement to methodology and results. Clear explanations of the hue- and saturation-equivariant group actions, along with detailed figures, aid in understanding the technical concepts. Comprehensive appendices further enhance reproducibility and transparency by providing proofs and additional experiment details.

**Weaknesses:**

1, The paper extends GCNNs to be equivariant to hue and saturation but does not address luminance variations, which are also important for color representation. Although the authors suggest that luminance invariance may be approximately preserved, this may not hold in real-world applications, such as medical imaging or scenes with varying lighting. Future work could integrate luminance into equivariant transformations or evaluate its impact on specific tasks.

2, The model is compared to standard CNNs (e.g., ResNet) and color-equivariant networks (e.g., CEConv), but not to other common approaches for handling color variations, such as color augmentation techniques or robust contrastive learning models. Adding these comparisons would strengthen the case for this method’s effectiveness in handling color shifts.

**Questions:**

1, The additional computational load from group convolutions and lifting layers present a challenge for real-time applications. Therefore, it is essential to compare your method with common alternatives, such as color augmentation or robust contrastive learning approaches, which are known to enhance robustness to color and perceptual variations. These comparisons would be especially valuable for demonstrating the unique advantages of color equivariance over standard augmentation techniques.

---

> ### Author Response · Authors · 2024-11-21
>
> > The authors do not not address luminance variations.
>
> To address luminance variation, we developed a luminance equivariant GCNN and compared its performance to a conventional CNN architecture on the small NORB dataset [1]. The small NORB dataset consists of 48.6k 96x96 grayscale images captured under 6 different lighting conditions, 9 elevations, and 18 azimuths. We partitioned the dataset into a training set and two test sets; in each dataset two of the six lighting conditions were represented.  When we trained and tested on the same luminance conditions, the conventional CNN and our new luminance equivariant GCNN performed comparably, around 92% accuracy. However, when we trained and tested on different luminance conditions, the performance of the conventional CNN dropped to 64% while our luminance equivariant GCNN maintained an accuracy of 78%. Thank you for pointing out this opportunity to extend the notion of color equivariance to luminance, we will definitely include these results in our paper!
>
> [1] LeCun, Yann, Fu Jie Huang, and Leon Bottou. "Learning methods for generic object recognition with invariance to pose and lighting." Proceedings of the 2004 IEEE Computer Society Conference on Computer Vision and Pattern Recognition, 2004. CVPR 2004.. Vol. 2. IEEE, 2004.
>
> > Adding comparisons to common approaches for handling color variations would strengthen the paper.
>
> We ran additional experiments to compare the performance of our color equivariant approach to baselines that use data augmentation or grayscale input images and found that our color equivariant approach performed on-par or better than the baseline color invariant approaches. We have included results from these experiments in Table 4. We’d like to point out that even though all methods performed similarly on classification, our color equivariant architecture is more interpretable by construction, has higher sample efficiency (see Figure 4 in our paper, and [2,3,4,5] who highlighting the sample efficiency of equivariant representation learning both empirically and theoretically), and naturally allows for downstream tasks such as color based sorting (see Figure 7) and color alignment.
>
> While we did not compare against robust contrastive learning approaches (e.g., [6]), we suspect they would show similar results on our benchmarks and, since they are color invariant by construction, they would have the limitations of color invariant models described above.
>
> [2] Behboodi, Arash, Gabriele Cesa, and Taco S. Cohen. "A pac-bayesian generalization bound for equivariant networks." Advances in Neural Information Processing Systems 35 (2022): 5654-5668.
> [3] Elesedy, Bryn. "Group symmetry in pac learning." ICLR 2022 workshop on geometrical and topological representation learning. 2022.
> [4] Zhu, Sicheng, Bang An, and Furong Huang. "Understanding the generalization benefit of model invariance from a data perspective." Advances in Neural Information Processing Systems 34 (2021): 4328-4341.
> [5] Brehmer, Johann, et al. "Does equivariance matter at scale?." arXiv preprint arXiv:2410.23179 (2024).
> [6] Lo, Yi-Chen, et al. "Clcc: Contrastive learning for color constancy." Proceedings of the IEEE/CVF Conference on Computer Vision and Pattern Recognition. 2021.

---

> > ### Comment · Reviewer_WTM5 · 2024-12-03
> >
> > Thanks for the author's response and my concerns have been addressed.

---

### Official Review · Reviewer_jSPt · 2024-11-02

**Soundness:** 4
**Presentation:** 3
**Contribution:** 3
**Rating:** 8
**Confidence:** 4

**Summary:**

The paper extends the concept of color equivariance to include saturation shifts, in addition to hue transformations, then they design GCNNs that are equivariant to color variation by leveraging the geometric structure of color in the hue-saturation-luminance (HSL) color space, additionally a novel lifting layer is proposed to transform input images directly, avoiding invalid RGB values and significantly improving equivariance error.

**Strengths:**

1. The paper is very well written and easy to read.

2. The authors conducted extensive experiments using natural images (CIFAR10, CIFAR100, STL-10, etc.) as well as medical images (Camelyon-17) to validate the effectiveness of their algorithm. The experiments are thorough and detailed.

3. Color equivariant representations are an important concept in computer vision, yet they are often overlooked in contemporary research. The authors effectively claim the significance of color in the era of deep learning in this work.

**Weaknesses:**

I believe this work has no significant drawbacks and meets the acceptance standards for ICLR.

Beyond that, I am curious about two questions:

1. This paper uses the HSL color space as the basis for designing experiments, specifically designing groups and group actions for hue and saturation, as well as a hue-saturation group action. I am interested to know whether this approach would also be applicable to other chromacity-luminance color spaces, such as CIELAB or YUV.  Additionally, the experiments are designed around hue shifts in the HSI series (i.e. H in HSV. HSL, HSB...). Could other chromacity shifts in different color spaces be designed to further demonstrate the robustness of the model?

2. The experiments presented in this paper focus on classification tasks. I wonder if this design could be even more beneficial for higher-level tasks or those that rely more on semantic information, such as semantic segmentation. Do the authors have any plans to design similar experiments for such tasks?

**Questions:**

See weakness part.

---

> ### Author Response · Authors · 2024-11-21
>
> > Would this approach also be applicable to other chromacity-luminance color spaces, such as CIELAB or YUV?
>
> We think it would be possible to approximate the structure of each of the three channels of the YUV color space using the group $(\mathbb{Z}, +)$. This is the same approach we used for the saturation (Section 4: Saturation group and group action) and luminance (Appendix E.1: Luminance group and group action) space in our proposed architecture. How well this would work, however, depends on how approximation errors accumulate. To model HSL, we only have to leverage the approximation twice. To model YUV, we would need to leverage the approximation three times.
>
> It’s not immediately clear to us if there are groups that capture the structure of the CIELAB color space but we will continue to think about possibilities.
>
> > Could other chromacity shifts in different color spaces be designed to further demonstrate the robustness of the model?
>
> Yes! As long as there is a mapping between the HSL and the other color space, our network should maintain robustness to transformations of the input.
>
> > I wonder if this design could be beneficial for semantic segmentation.
>
> That is an exciting direction we had not thought of! Semantic segmentation might benefit from the principled way our network can organize and discard color information. Our network might also be useful for generating superpixels that can be used in downstream processes. Since our design can be used with any CNN backbone, it would be interesting to see if incorporating it into an existing pipeline yields any benefit.

---

> > ### Comment · Reviewer_jSPt · 2024-11-26
> >
> > Thanks for the author's response, I'll maintain my original rating.

---

### Official Review · Reviewer_y47K · 2024-11-05

**Soundness:** 2
**Presentation:** 3
**Contribution:** 2
**Rating:** 6
**Confidence:** 2

**Summary:**

This paper introduces group convolutional neural networks designed to be equivariant to variations in hue, saturation, and color in RGB images. It proposes a lifting layer that directly transforms the input image to produce a color-equivariant descriptor, effectively addressing the challenge of invalid RGB values found in previous works. Experiments on synthetic and real-world datasets highlight the robustness of the proposed method to color variations.

**Strengths:**

1.	The approach is technically compelling, as it enables convolutional neural networks to be equivariant to hue and saturation by design, leveraging the inherent geometric structure of these color components.

2.	Results across several benchmarks indicate that the proposed method offers stronger robustness to color variation compared to existing methods.

3.	The paper is well-structured and easy to understand.

**Weaknesses:**

1.	Experiments are limited to small toy datasets with pronounced hue or saturation shifts, leaving it unclear how the method would perform on large datasets like ImageNet.

2.	The proposed method focuses on hue and saturation equivariance, but does not address other common variations in real-world data, such as 3D geometric transformations or lighting changes.

**Questions:**

Would the proposed method improve image classification performance on large-scale datasets like ImageNet or on tasks beyond image classification?

**Details Of Ethics Concerns:**

No ethics concerns

---

> ### Author Response · Authors · 2024-11-21
>
> > How would the method perform on large datasets like ImageNet?
>
> To see how our approach compares to baseline models on a larger dataset, we trained our model (Hue-4), ResNet-18, and CEConv-4 on the Tiny-ImageNet dataset [1] and got a classification error (lower is better) of 52.95%, 53,62%, and 54.50% respectively. These results suggest our approach performs on-par with conventional architectures on larger datasets. The Tiny-ImageNet dataset is a downsized subset of ImageNet consisting of 100k 64x64 RGB images. Of the 1000 classes represented in ImageNet, 200 are represented in Tiny-ImageNet, each with 500 training examples, 50 validation examples, and 50 test examples. We chose to perform our experiments on Tiny-ImageNet due to time constraints. We will include results on the full ImageNet dataset in the camera ready version.
>
> [1] Le, Ya, and Xuan Yang. "Tiny imagenet visual recognition challenge." CS 231N 7.7 (2015): 3.
>
> > Other common variations in real-world data, such as 3D geometric transformations or lighting changes are not addressed.
>
> 3D Geometric transformations such as 3D rotations are important real-world variations and are the focus of equivariant representation learning approaches such as [2,3,4,5]. In our paper, we exclusively consider color variation which has been shown to adversely affect predictive performance [6], and is significantly less explored (as far as we know, only [7] has explored equivariant representation learning in the context of color).
>
> Luminance variation is also important in real-world applications. We developed a luminance equivariant GCNN and compared its performance to a conventional CNN architecture on the small NORB dataset [8]. The small NORB dataset consists of 48.6k 96x96 grayscale images captured under 6 different lighting conditions, 9 elevations, and 18 azimuths. We partitioned the dataset into a training set and two test sets; in each dataset two of the six lighting conditions were represented. When we trained and tested on the same luminance conditions, the conventional CNN and our new luminance equivariant GCNN performed comparably, around 92% accuracy. However, when we trained and tested on different luminance conditions, the performance of the conventional CNN dropped to 64% while our luminance equivariant GCNN maintained an accuracy of 78%. Thank you for pointing out this opportunity to extend the notion of color equivariance to luminance, we will definitely include these results in our paper!
>
> [2] Esteves, Carlos, et al. "Learning so (3) equivariant representations with spherical cnns." Proceedings of the European Conference on Computer Vision (ECCV). 2018.
> [3] Cohen, Taco S., et al. "Spherical CNNs." International Conference on Learning Representations. 2018.
> [4] Thomas, Nathaniel, et al. "Tensor field networks: Rotation-and translation-equivariant neural networks for 3d point clouds." arXiv preprint arXiv:1802.08219 (2018).
> [5] Fuchs, Fabian, et al. "Se (3)-transformers: 3d roto-translation equivariant attention networks." Advances in neural information processing systems 33 (2020): 1970-1981.
> [6] De, Kanjar, and Marius Pedersen. "Impact of colour on robustness of deep neural networks." Proceedings of the IEEE/CVF international conference on computer vision. 2021.
> [7] Lengyel, Attila, et al. "Color equivariant convolutional networks." Advances in Neural Information Processing Systems 36 (2024).
> [8] LeCun, Yann, Fu Jie Huang, and Leon Bottou. "Learning methods for generic object recognition with invariance to pose and lighting." Proceedings of the 2004 IEEE Computer Society Conference on Computer Vision and Pattern Recognition, 2004. CVPR 2004.. Vol. 2. IEEE, 2004.

---

> > ### Comment · Reviewer_y47K · 2024-11-26
> >
> > Thank you very much for the detailed response! I decide to raise my rating from 5 to 6. The scope of this work is relatively small as it focuses on the color equivariance while there are so many other aspects of variations in the real world. Would we design such framework for each variation (e.g. 3D geometric transformations, color, texture, depth, etc.)? It is hard to convince myself this is a scalable approach and a promising research direction. However, I totally understand this is more about my personal research taste. I still enjoy reading works that provide elegant formulations.

---

### Author Response · Authors · 2024-11-21

We thank the reviewers for their thoughtful feedback. We are encouraged they found our work to be well motivated (7Wsa, r9ZC); technically compelling/novel/inventive (y47, jSPt, WTM5); effective in addressing the challenge of invalid RGB values (y47, jSPt, WTM5); and robust to color variation compared to existing methods (y47, WTM5, 7Wsa). We are also glad they found our claims to be well substantiated by the numerous experiments (y47, jSPt, WTM5, 7Wsa) and analyses (WTM5); and our paper to be well written, well illustrated, and easy to read (y47, jSPt, WTM5, 7Wsa, r9ZC).

Reviewers y47 and WTM5 noted the prevalence of luminance variation in real-world data. To explore this, we evaluated the generalization performance of a conventional CNN and **our newly developed luminance equivariant GCNN** on the small NORB dataset [1]. The small NORB dataset consists of 48.6k 96x96 grayscale images captured under 6 different lighting conditions, 9 elevations, and 18 azimuths. When we trained and tested on the same luminance conditions, the conventional CNN and our new luminance equivariant GCNN performed comparably, around 92% accuracy. However, when we trained and tested on different luminance conditions, the performance of the conventional CNN dropped to 64% while our luminance equivariant GCNN maintained an accuracy of 78%. The details of how we define our luminance space, and luminance transformation are provided in Appendix E.1. Thank you for pointing out this opportunity to extend the notion of color equivariance to luminance, we will definitely include these results in our paper!

Reviewers WTM5 and R9ZC suggested comparisons against baselines that use data augmentation or grayscale input images. **We ran additional experiments to compare the performance of our color equivariant approach to the suggested color invariant approaches** and found that our color equivariant approach performed on-par or better than the suggested color invariant approaches. We have included these results in Table 4. We’d like to point out that even though all methods perform similarly on classification, our color equivariant architecture is more interpretable by construction, has higher sample efficiency (which others have shown both theoretically and empirically [2,3,4,5], and we’ve shown Figure 4), and naturally allows for downstream tasks such as color based sorting (see Figure 7) and color alignment.

We address other reviewer comments below and will incorporate all feedback. We highlight changes to our original manuscript with blue text.

[1] LeCun, Yann, Fu Jie Huang, and Leon Bottou. "Learning methods for generic object recognition with invariance to pose and lighting." Proceedings of the 2004 IEEE Computer Society Conference on Computer Vision and Pattern Recognition, 2004. CVPR 2004.. Vol. 2. IEEE, 2004.
[2] Behboodi, Arash, Gabriele Cesa, and Taco S. Cohen. "A pac-bayesian generalization bound for equivariant networks." Advances in Neural Information Processing Systems 35 (2022): 5654-5668.
[3] Elesedy, Bryn. "Group symmetry in pac learning." ICLR 2022 workshop on geometrical and topological representation learning. 2022.
[4] Zhu, Sicheng, Bang An, and Furong Huang. "Understanding the generalization benefit of model invariance from a data perspective." Advances in Neural Information Processing Systems 34 (2021): 4328-4341.
[5] Brehmer, Johann, et al. "Does equivariance matter at scale?." arXiv preprint arXiv:2410.23179 (2024).

---

> ### Author Response · Authors · 2024-11-25
> **New foreground-background segmentation results**
>
> Reviewers jSPt suggested that color-equivariance might benefit semantic segmentation. To explore this, we compared the performance of a conventional CNN based foreground-background segmentation architecture with a hue-equivariant variant on the Caltech-101 dataset [1]. We found that our hue-equivariant architecture outperforms the conventional architecture with an accuracy of 90.33 (0.23) compared to 88.75 (0.12), and produces foreground-background segmentation masks that are more compact, have fewer artifacts, and better capture fine details (see Figure 20). The architecture details, and quantitative and qualitative results are provided in Appendix E.3. Thank you for suggesting this application of our method, we will definitely include these results in our paper!
>
> [1] Fei-Fei, Li, Rob Fergus, and Pietro Perona. "Learning generative visual models from few training examples: An incremental bayesian approach tested on 101 object categories." 2004 conference on computer vision and pattern recognition workshop. IEEE, 2004.

---

### Meta-Review · Area_Chair_tdub · 2024-12-19

**Metareview:**

This submission introduces a colour-equivariant group convolutional network (GCNN) that generalises equivariance beyond the hue channel to saturation in RGB images. The proposed method employs a lifting layer capable of directly transforming input images and further leverages the geometric structure in hue-saturation-luminance (HSL) colour space. Experiments on synthetic and real-world datasets show improved classification performance over baseline approaches.

This paper was discussed at length with the SAC. After reviewing the paper, rebuttal and resulting discussions AC believes that the overall strengths outweigh the weaknesses and recommends acceptance. For the camera-ready version, the authors should incorporate all key rebuttal results and discussion points as suggested.

**Additional Comments On Reviewer Discussion:**

The paper received five reviews resulting in: one strong accept, two borderline accepts, one borderline reject and one clear reject.

Reviewers noted positive aspects relating to (i) technically compelling; (ii) thorough and extensive experimental work; (iii) well written and easy to follow. Negative, and in some cases contrary, review comments raised concerns pertaining to narrow scope with respect to real-world variance, limited and insufficient comparisons with baselines, overclaims regarding empirical improvements, lack of technical details, and marginal performance gains. Suggestions for extension included exploring additional colour spaces & luminance, large-scale data and additional down-stream tasks.

The authors performed well in the rebuttal and could resolve a set of reviewer concerns by presenting additional results and explanations which AC notes can meaningfully improve the manuscript. Post-rebuttal one reviewer raised their score and others explicitly state that the submission meets the acceptance standards. A further reviewer remained unconvinced citing unresolved concerns relating to experimental results (Table 1, 2, and 4), lack of clear understanding for reported observations and overclaims on sample efficiency.

---

### Decision · Program_Chairs · 2025-01-22

Accept (Poster)